# ECHO: Elastic Speculative Decoding with Sparse Gating for High-Concurrency Scenarios

Xinyi Hu [* 1]  Yuhao Shen [* 2 1]  Baolin Zhang [* 1]  Hengxin Zhang [1]  Jun Dai [1]  Shuang Ge [† 1]  Lei Chen [1]  Yue Li [1]  Mingcheng Wan [1]

## Abstract

Speculative Decoding promises to accelerate the inference of Large Language Models, yet its efficacy often degrades in production-grade serving. Existing evaluations typically overlook the compute-bound nature of high-concurrency regimes, where verification compute becomes the dominant bottleneck. Consequently, prior methods face a dilemma: static trees incur massive verification waste, while dynamic trees suffer from cumulative misjudgments and kernel incompatibility. To bridge this gap, we introduce *ECHO*, a high-concurrency-oriented framework integrated into SGLang that reformulates speculative execution as a budgeted scheduling problem. Crucially, *ECHO* employs sparse confidence gating to manage the batch as a unified supertree, elastically pivoting budget between depth and width to co-optimize the trade-off between reducing global verification steps and maximizing per-step efficiency. Extensive evaluations across diverse model scales–particularly the industrial-grade Qwen3-235B–demonstrate that *ECHO* consistently outperforms SOTA methods in both low-load and high-load scenarios, achieving up to 5.35× wall-time speedup and delivering over 20% relative speedup gain.

## 1. Introduction

As Large Language Models (LLMs) scale toward massive parameter counts, high-concurrency, and long-context scenarios, the inefficiency of autoregressive (AR) decoding becomes a first-order bottleneck for production-level systems (Yang et al., 2025; Guo et al., 2025; Singh et al., 2025;

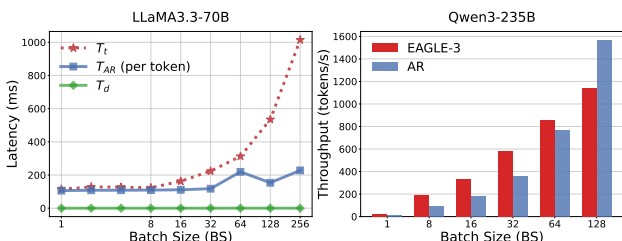

*Figure 1.* **Performance degradation in high-concurrency scenarios on MT-Bench. Left:** Latency breakdown for LLaMA-3.3-70B. Verification cost ($T_t$) scales linearly, becoming the dominant bottleneck over AR cost ($T_{AR}$) as batch size increases. **Right:** Throughput for Qwen3-235B. Due to this bottleneck, the speedup of EAGLE-3 diminishes and eventually underperforms vanilla AR at high concurrency (BS=128).

Google DeepMind Team, 2025; Sadhukhan et al., 2024).

Speculative Decoding (SD) mitigates this serial barrier via a draft then verify paradigm (Leviathan et al., 2023; Chen et al., 2023): a lightweight draft model predicts multiple tokens ahead, and the target model verifies them in parallel, amortizing expensive target computation across multiple accepted tokens. Tree-based drafting further enlarges the candidate set within each verification step (Li et al., 2025b; Miao et al., 2024), often improving the accepted length per step in low-concurrency settings. However, we observe that these gains degrade precipitously in high-concurrency scenarios, as shown in Figure 1.

This degradation is not incidental: modern serving operates in an increasingly compute-bound regime, where many requests contend for the target model's verification compute, and any wasted verified token directly translates into lost goodput and worse tail latency. This motivates a broader question beyond any single tree heuristic:

*How can we establish a principled methodology for SD in LLM serving, where verification compute is the primary bottleneck and system-level constraints dictate whether algorithmic efficiency translates into actual goodput gains?*

Revisiting prior work reveals a core conflict in high-concurrency scenarios: the trade-off between the verification waste of static methods and the cost of misjudgment in dynamic methods. Static tree methods (Miao et al., 2024; Li

---
[*]Equal contribution [1]Qwen Applications Business Group of Alibaba, Hangzhou, China [2]Zhejiang University, Hangzhou, China. Correspondence to: Shuang Ge <geshuang.zj@alibaba-inc.com>.

*Proceedings of the 43rd International Conference on Machine Learning*, Seoul, South Korea. PMLR 306, 2026. Copyright 2026 by the author(s).

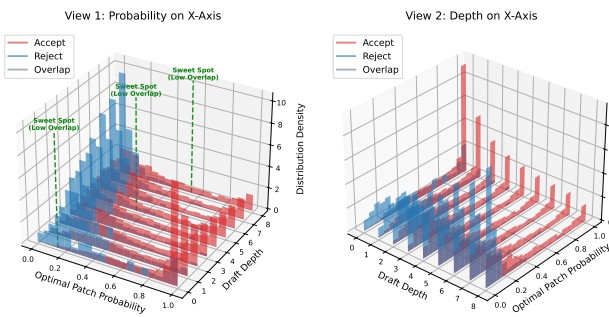

*Figure 2.* **Visualization of confidence distributions across draft depths** (LLaMA3.1-8B on MT-Bench). The two views show the probability of accepted and rejected draft tokens at different tree depths. Low overlap indicates better discriminability of the confidence scores, so these layers are identified as "Sweet Spots".

et al., 2024b; 2025b) use fixed tree structures. While simple, they often verify too many useless branches when the model is uncertain. This leads to significant waste: although they aim to reduce the total number of steps, the large tree size makes each single verification step overly time-consuming. Dynamic tree strategies (Liu et al., 2026; Brown et al., 2024) try to reduce this waste by adjusting the tree online, but they face severe challenges in production. First, prediction errors often accumulate in high-concurrency settings, canceling out the expected speed gains. Second, they create irregular batch shapes, which are not supported by modern SD frameworks. System schedulers (Wu et al., 2025; Liu et al., 2025a) manage global budgets but typically treat the draft tree as a "black box," ignoring the inefficiency within the tree construction itself. In short, the community still lacks a production-ready solution that minimizes both verification waste and the extra costs of dynamic control.

It becomes clear that effective SD in production should take care of two factors: (1) **reducing global verification steps**, and (2) **increasing per-step verification efficiency**. Prior methods typically optimize one metric at the expense of the other, lacking a precise mechanism to reconcile this delicate trade-off. To resolve this tension, we first investigate why the per-step efficiency gains from dynamic adjustment fail to offset the overhead of increased verification steps. We identify the root cause as the frequent misjudgment present in current expansion policies. As illustrated in Figure 2, the discriminative power of confidence scores is inherently depth-dependent: the root and target depth consistently act as high-fidelity "sweet spots" with sharp signal-to-noise ratios, whereas most intermediate layers exhibit blurred decision boundaries. Consequently, prior work (Huang et al., 2025; Li et al., 2025c; Hong et al., 2025) compels decisions at noisy intermediate layers, introducing cumulative misjudgment risks that negate potential gains.

Driven by this insight, we present *ECHO* (*Elastic Speculative Decoding with Sparse Gating for High-Concurrency Scenarios*), a framework that reformulates tree construction

as a budget scheduling problem. To align with the requirements of high-concurrency scenarios, *ECHO* operates under a strict fixed verification budget. Within this fixed envelope, *ECHO* employs a sparse gating strategy anchored on high-fidelity "sweet spots"–primarily the root and target depth, supplemented by a few discriminative intermediate layers selected dynamically. Specifically, we trigger a "Truncate-to-Widen" operation at these checkpoints to maximize yield under uncertainty, or an "Extend" operation to capitalize on momentum. **This selective elasticity enables *ECHO* to aggressively deepen the candidate tree to reduce global verification steps when confidence is high, while confidence is insufficient, pivoting to width to safeguard per-step efficiency.**

Crucially, we implement *ECHO* atop the industrial-grade SGLang framework. Unlike prior dynamic tree methods that primarily report results on raw transformers—leaving their incompatibility with standard serving kernels unaddressed—we explicitly implemented specialized operators to support irregular batch shapes within SGLang. This system-level adaptation allows the tree structure to fluidly morph to match the request's entropy, effectively bridging the gap between theoretical dynamic algorithms and production deployment.

Our main contributions are as follows:

- We formulate a serving-centric methodology for SD in modern LLM serving, highlighting verification compute as the dominant scarce resource and identifying two primary failure modes under system constraints: verification waste and misjudgment accumulation in dynamic control.

- We propose *ECHO*, a training-free framework that employs sparse gating at high-fidelity "sweet spots" (root, target depth, and adaptive intermediate layers). By scheduling width vs. depth under a fixed budget, *ECHO* minimizes misjudgment accumulation.

- We validate *ECHO* on various models, especially the large-scale Qwen3-235B. The results demonstrate that *ECHO* consistently outperforms state-of-the-art (SOTA) methods, particularly in speculation-hard regimes where prior art degrades.

- We provide the implementation of dynamic-tree SD for high-concurrency scenarios and integrated it into SGLang for the first time, bridging the gap between academic research and production deployment.

## 2. Problem Formulation

**Standard SD.** SD pairs a lightweight draft model with a target model. At step $t$ with prefix $x_{1:t}$, the draft model proposes $K$ draft tokens, and the target verifies them in one parallel forward pass, accepting a prefix of length $L \in$

$\{0, \ldots, K\}$ (e.g., via rejection sampling (Leviathan et al., 2023)).

A common Speedup proxy is

$$Speedup = \frac{(\mathbb{E}[L] + 1)\, T_{\mathrm{ar}}}{T_{\mathrm{draft}}(K) + T_{\mathrm{verify}}(K)}, \qquad (1)$$

where $\mathbb{E}[L]$ represents the mean accepted tokens (MAT), $T_{\mathrm{ar}}$ is the cost of one autoregressive step, and $T_{\mathrm{draft}}/T_{\mathrm{verify}}$ correspond to the drafting and verification latency. Prior work often assumes $T_{\mathrm{verify}} \approx T_{\mathrm{ar}}$ for moderate $K$, motivating the use of large draft token trees to maximize $\mathbb{E}[L]$ regardless of the verification overhead (Li et al., 2025b).

**SD in High-Concurrency Scenarios.** In production serving, a batch contains $B$ concurrent requests. Let request $i$ propose $K_i$ tokens. The total number of tokens to verify in one step is $K_{\mathrm{total}} = \sum_{i=1}^{B} K_i$. Recent studies (Liu et al., 2025b; Sadhukhan et al., 2024) show that verification becomes **compute-bound** when $K_{\mathrm{total}}$ is large. This often happens with large models and long contexts. In this case, the latency grows linearly with the number of tokens:

$$T_{\mathrm{ver}}(K_{\mathrm{total}}) \approx T_{\mathrm{ar}}(1 + \gamma \cdot [K_{\mathrm{total}} - K_{\mathrm{max}}]^+), \qquad (2)$$

where $K_{\mathrm{max}}$ is the computing limit of the hardware. This means the "free-lunch" assumption fails: once the total load exceeds $K_{\mathrm{max}}$, any increase in $K_i$ linearly penalizes the verification latency for all concurrent requests.

**Budget-Constrained Objective.** In compute-bound regimes, SD is best viewed as a verification-budget allocation problem. Under a strict budget, the objective is no longer solely to maximize $\mathbb{E}[L]$ via aggressive tree expansion (aiming to reduce global verification steps), but to maximize per-step verification efficiency. We quantify this by draft Yield:

$$\mathtt{Yield} = \frac{\mathbb{E}[L]}{1 + [K_{\mathrm{total}} - K_{\mathrm{max}}]^+}, \qquad (3)$$

i.e., the fraction of verified tokens that contribute to the accepted prefix. Rigid static trees may achieve high $\mathbb{E}[L]$ but incur large verification waste via low-utility branches, while dense dynamic control can introduce decision errors and overhead that accumulate under concurrency. This motivates a serving-oriented design that improves Yield under a fixed verification budget with minimal control.

## 3. Method

### 3.1. Overview: The *ECHO* Framework

We propose *ECHO*, a budget-aware speculative decoding framework that unifies tree construction (depth vs. width)

and batch-level scheduling (request-to-request allocation) under a single serving-centric objective: maximize useful progress under a fixed verification compute budget. In modern high-concurrency scenarios, verification easily becomes compute-bound, leaving a narrow margin for speedup. Consequently, the system is strictly constrained by the total number of tokens verified in each step. *ECHO* therefore models each verification iteration as a global token-budget allocation problem. Figure 3 illustrates the unified pipeline, showing how the unified scheduler coordinates dynamic tree construction and elastic resource reallocation before packing the batch for efficient execution.

**Super-Tree View.** Consider a batch of $B$ concurrent requests. At a given SD iteration, each request $i$ constructs a candidate token tree $\mathcal{G}_i$. Let $\mathcal{V}_i$ denote the set of all candidate nodes in $\mathcal{G}_i$. The target model verifies the flattened union $\bigcup_{i=1}^{B} \mathcal{V}_i$ in one parallel forward pass (Li et al., 2024a).

We impose a batch-level verification cap:

$$\sum_{i=1}^{B} K_i \leq K_{\mathrm{max}}, \qquad K_i \triangleq |\mathcal{G}_i|, \qquad (4)$$

where $K_i$ is the number of candidate tokens submitted for request $i$, and $K_{\mathrm{max}}$ is set near the hardware saturation threshold (i.e., the compute-bound limit). Under this view, the batch behaves as a single super-tree sharing one global budget $K_{\mathrm{max}}$: allocating additional candidates to one request necessarily reduces the candidates available to others. In practice, $K_{\mathrm{max}}$ is determined once at engine startup by sweeping static-tree batch sizes and locating the memory-to-compute-bound latency inflection point. This makes the cap hardware-, model-, and parallelism-aware without manual retuning for each batch size.

**Core Components.** *ECHO* couples two key components that jointly determine where and how to spend the budget:

- **Sparse Confidence Gating:** A lightweight reliability signal evaluated only at a sparse set of discriminative "sweet spots" (e.g., root, target depth, and select intermediate depths). This avoids the overhead and misjudgment accumulation of per-depth/node control.

- **Elastic Budget Scheduling:** A unified scheduler that reallocates the shared budget across two axes: (1) **within a request** (adjusting depth vs. width), and (2) **across requests** (shifting budget from one request to another), all constrained by Eq. 4.

**Kernel-compatible execution.** The scheduler produces ragged per-request trees, but the SGLang integration does not rely on padding. Instead, *ECHO* performs sequence-level truncation on input_ids, then flattens and packs only the

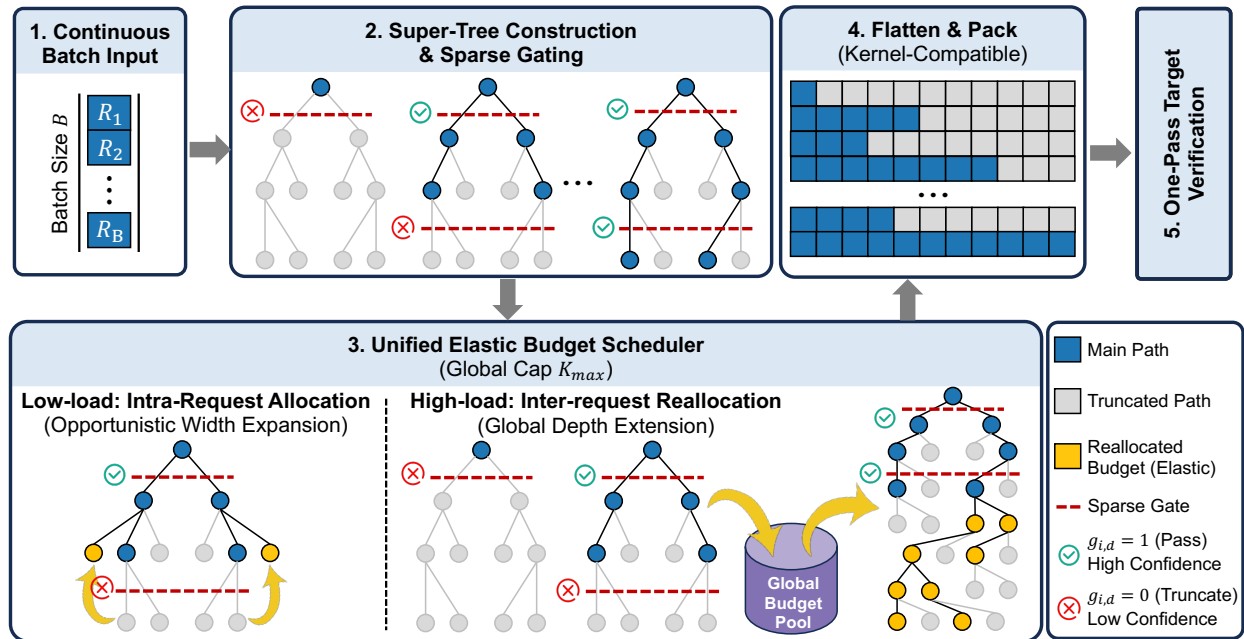

*Figure 3.* **Overview of the *ECHO* Framework. (1) Super-Tree Construction:** Draft trees are evaluated (truncated or extended) only at sparse gates. **(2) Unified Elastic Budget Scheduler:** Under a global verification cap ($K_{\max}$), the scheduler dynamically adapts resource allocation. In Low-Load scenarios, the budget saved by truncation is reused locally to **widen** the current tree. In High-Load scenarios, budget saved from truncated low-confidence requests is reallocated to extend the **depth** of high-confidence requests. **(3) Flatten & Pack:** Finally, the ragged batch formed by requests with varying token counts is packed into a dense, kernel-compatible layout for efficient verification.

surviving candidate tokens for verification. As a result, both Attention and GEMM work decrease with the pruning ratio. The required control metadata–confidence scores, topology offsets, and budget states–scales as $O(K_{\max})$ and is only tens of KB per batch, which is negligible relative to the GB-scale KV cache in the evaluated serving setup.

### 3.2. Sparse Gating at Sweet Spots

Prior dynamic tree methods(Liu et al., 2026; Brown et al., 2024) often perform dense, fine-grained control (e.g., per-depth/node) to adjust topology. However, our analysis indicates that the discriminative power of confidence scores is depth-dependent (Figure 2): only a few depths act as reliable "sweet spots", while most depths exhibit blurred decision boundaries. *ECHO* exploits this structure by restricting gating to a sparse set of checkpoint depths.

**Path Scoring.** Consider the draft tree for request $i$. For a node $j \in \mathcal{V}_{i,d}$ at depth $d$ corresponding to token $x_{i,d,j}$, we define its path score (cumulative log-score) recursively:

$$S_{i,d,j} = S_{i,d-1,\mathrm{pa}(j)} + \log q\big(x_{i,d,j} \mid h_{i,d-1,\mathrm{pa}(j)}\big), \quad (5)$$

where $\mathrm{pa}(j)$ denotes the parent node index, $h_{i,d-1,\mathrm{pa}(j)}$ is the hidden state used to generate the current token.

**Layer Confidence.** We summarize the reliability of the entire candidate set $\mathcal{V}_{i,d}$ at depth $d$ for request $i$ using the maximum-likelihood path probability:

$$c_{i,d} = \exp\left( \max_{j \in \mathcal{V}_{i,d}} S_{i,d,j} \right). \quad (6)$$

Intuitively, $c_{i,d}$ measures whether there exists at least one highly probable path up to depth $d$. This layer-level summary also defines how *ECHO* handles branches within the same depth: it uses the best cumulative path score to decide whether the depth is reliable, rather than maintaining separate per-branch gates. This avoids dense branch tracking and keeps the control path compatible with high-concurrency serving.

**Offline Calibration.** Similarly to the calibration phase in Post-Training Quantization (Xiao et al., 2023; Lin et al., 2024), *ECHO* determines the gating parameters during the warm-up period with almost no additional overhead. We quantify the separability of confidence scores $c_{i,d}$ for accepted versus rejected tokens using the AUC metric (Hanley & McNeil, 1982). We identify "sweet spots" $\mathcal{D}_{\mathrm{sig}}$ as depths exhibiting high discriminative power (i.e., $\mathrm{AUC}_d > \delta$) and calibrate the thresholds $\tau_d$ at these checkpoints to maximize the separation between the two distributions. This calibration is lightweight: it reuses the first few draft steps to estimate the confidence thresholds, rather than requiring a separate profiling run. Since the thresholds are tied to the draft model's confidence distribution, they are not retuned

for each batch size or serving workload. Appendix D shows that thresholds calibrated on HumanEval transfer to other tasks with only 1–3% throughput degradation.

**Gating Signal.** During inference, *ECHO* restricts gating decisions strictly to the pre-calibrated checkpoints $d \in \mathcal{D}_{\text{sig}}$. At these depths, the system emits a binary control signal based on the learned thresholds:

$$g_{i,d} = \mathbb{I}\left[c_{i,d} > \tau_d\right]. \tag{7}$$

Here, $g_{i,d} = 1$ signals high confidence (safe to extend depth), while $g_{i,d} = 0$ signals low confidence (high probability of rejection), triggering the scheduler to truncate and reallocate budget.

### 3.3. Unified Elastic Budget Scheduling

The core of *ECHO* treats the verification cap $K_{\text{max}}$ as a shared global pool. To ensure fair resource competition, the scheduler operates depth-by-depth across the batch, applying a unified priority rule to resolve the key allocation trade-off: whether to reinvest locally by widening the current draft tree at the truncation depth, or to reallocate globally to support other high-potential requests by allocating a larger token budget.

**Priority-Based Allocation.** At decision step, we allocate the remaining budget based on a strict priority hierarchy:

- **Priority 1: Global Depth Extension.** This is the primary objective, **aimed at aggressively reducing the total verification steps**. As long as any request in the batch maintains high confidence ($g_{i,d} = 1$), the global budget is prioritized for its depth extension. If a request is truncated ($g_{i,d} = 0$), it yields the budget to other active requests ($j \neq i$) that are still deepening.

- **Priority 2: Opportunistic Width Expansion.** This is triggered only when no active requests can further extend depth (e.g., all requests are truncated). Only then is the surplus budget reallocated to widen the candidate set of the truncated requests to improve coverage.

In a resource-contended batch, the budget is reserved for the depth expansion of confident requests before any width expansion is permitted.

**Elastic Scheduling.** The unified priority rule adapts naturally to different load conditions (Alg. 1):

- **Case 1 (Low-load: Intra-request Allocation).** When the budget is ample (e.g., $B = 1$ or low contention), Priority 1 cannot fully absorb the available capacity. The scheduler naturally falls back to Priority 2, reinvesting the surplus locally to widen the draft tree at the truncation depth. This "Truncate-and-Widen" behavior enhances robustness when depth extension is risky.

---

**Algorithm 1** *ECHO*: Unified Elastic Budget Scheduling

1: **Input:** Batch $B$, Budget $K_{\text{max}}$, Gating params $(\mathcal{D}_{\text{sig}}, \tau)$, Max Width $W_{\text{max}}$, Default Width $W_{\text{topk}}$
2: **Output:** Draft Trees $\{\mathcal{G}_i\}_{i=1}^{B}$
3: **Init:** $budget \leftarrow K_{\text{max}}$, $d \leftarrow 0$, $active\_set \leftarrow \{1..B\}$, $trunc\_set \leftarrow \emptyset$
4: {**Phase 1: Global Depth Extension**}
5: **while** $budget > 0$ **and** $active\_set \neq \emptyset$ **do**
6:     $d \leftarrow d + 1$
7:     **for** $i \in active\_set$ **if** $budget > 0$ **do**
8:         $is\_pass \leftarrow (c_{i,d} > \tau_d)$ **if** $d \in \mathcal{D}_{\text{sig}}$ **else** True
9:         **if** $is\_pass$ **then**
10:           $\mathcal{G}_i.append(topk\_tokens)$
11:           $budget \leftarrow budget - W_{\text{topk}}$
12:         **else**
13:           $active\_set.remove(i);$    $trunc\_set.add(i)$
14:         **end if**
15:     **end for**
16: **end while**
17: {**Phase 2: Opportunistic Width Expansion**}
18: **for** $i \in trunc\_set$ **if** $budget \geq k$ **do**
19:     $w \leftarrow \min(budget, W_{\text{max}})$
20:     $\mathcal{G}_i.widen(k);$    $budget \leftarrow budget - W_{\text{max}}$
21: **end for**

---

- **Case 2 (High-load: Inter-request Reallocation).** When the batch is saturated (high contention), Priority 1 dominates. A truncated request ($g_{i,d} = 0$) yields its potential budget to the global pool, where it is immediately claimed for the depth extension of other high-confidence requests. This mechanism effectively reallocates compute from low-confidence to high-potential requests to maximize aggregate throughput.

## 4. Theoretical Motivation

We summarize two simple formal results that motivate *ECHO*'s design; full derivations are deferred to Appendix A.

**Width improves coverage at a truncated depth.** When depth extension is halted at depth $d$ (low confidence), *ECHO* uses remaining budget to widen the frontier. The next theorem states that widening strictly increases the probability of covering the target token at that depth.

**Theorem 1** (Coverage Gain via Width). *Let $\mathcal{S}_k = \{x^{(1)}, \ldots, x^{(k)}\}$ be the top-$k$ candidate tokens at depth $d$ ranked by the target next-token distribution $p_t(x \mid x_{<d})$. Let $x^*$ be the ground-truth token sampled from $p_t$, and define the coverage probability as $\mathbb{P}(x^* \in \mathcal{S}) \triangleq \sum_{x \in \mathcal{S}} p_t(x)$. If we expand from $k$ to $k' > k$, then the coverage probability*

*increases by the added probability mass:*

$$\mathbb{P}(x^* \in \mathcal{S}_{k'}) - \mathbb{P}(x^* \in \mathcal{S}_k) = \sum_{i=k+1}^{k'} p_t(x^{(i)} \mid x_{<d}) \; > \; 0,$$

$$(8)$$

*whenever $\sum_{i=k+1}^{k'} p_t(x^{(i)} \mid x_{<d}) > 0$.*

**Compute-bound serving favors reallocating budget to higher marginal utility.** Under saturated compute-bound serving, *ECHO* enforces a per-iteration token cap (Eq. 4) so that iteration time is dominated by verifying a fixed number of tokens. Thus, improving end-to-end throughput reduces to increasing the batch-level expected accepted tokens per iteration minimizing total verification steps, not necessarily each request's mean acceptance length. The next theorem formalizes a sufficient condition: moving budget from low marginal gain to high marginal gain strictly improves the batch objective.

**Theorem 2** (Marginal Utility Exchange). *Consider a batch iteration with fixed verification budget $\sum_{i=1}^{B} K_i = K_{\max}$. Let $f_i(k) \triangleq \mathbb{E}[L_i \mid K_i = k]$ be request $i$'s expected accepted tokens given $k$ verified candidates, and define the marginal gain $\Delta_i(k) \triangleq f_i(k) - f_i(k-1)$. For any two requests $i, j$, if*

$$\Delta_j(K_j + 1) \; > \; \Delta_i(K_i), \qquad (9)$$

*then reallocating one token from $i$ to $j$ (i.e., $K_i \leftarrow K_i - 1$, $K_j \leftarrow K_j + 1$) strictly increases the batch-level expected accepted tokens $\sum_{m=1}^{B} \mathbb{E}[L_m]$, and therefore improves end-to-end throughput in compute-bound serving.*

**From theory to implementation.** Theorem 2 describes the ideal allocation rule: under a fixed verification budget, tokens should move from requests with low marginal utility to requests with high marginal utility. In practice, the scheduler cannot observe the true marginal utility before target-model verification. *ECHO* therefore estimates it with the normalized path confidence score at the calibrated sweet-spot depths. Figure 2 supports this approximation: at these depths, confidence separates likely accepted tokens from likely rejected tokens. This proxy only affects speculative-budget allocation; correctness is unchanged because the target model still performs the final acceptance check.

## 5. Experiments

### 5.1. Experimental Setting

**Datasets and Models.** We evaluated *ECHO* in various LLMs, including Vicuna-13B (Chiang et al., 2023), LLaMA-3.1-8B, LLaMA-3.3-70B (Grattafiori et al., 2024), and the Qwen3 series (8B/32B/235B) (Yang et al., 2025). Following

protocols established by EAGLE (Li et al., 2024a) and Spec-Bench (Xia et al., 2024), our evaluation spans five comprehensive benchmarks covering code generation, mathematical reasoning, summarization, and chat: HumanEval (Chen et al., 2021), GSM8K (Cobbe et al., 2021), CNN/DM (Nallapati et al., 2016), Alpaca (Taori et al., 2023), and MT-Bench (Zheng et al., 2023).

**Baselines and Implementation.** We benchmark *ECHO* against representative methods across four distinct categories of SD: (1) Standard SD (Chen et al., 2023); (2) Retrieval-based (Fu et al., 2024); (3) Training-based (Cai et al., 2024; Ankner et al.; Li et al., 2025b); and (4) Dynamic Trees (Wang et al., 2025; Brown et al., 2024), which we adapted to support EAGLE-3 for fair comparison. Experiments are conducted on 8×H100 GPUs with greedy sampling. We use HuggingFace `transformers` for low-load (BS=1) and SGLang for high-load scenarios. All baselines are reproduced using their official configurations, with detailed hyperparameters provided in the Appendix C.

**Metrics** Since *ECHO* strictly adheres to the speculative sampling acceptance conditions and does not modify the target model's weights, the output distribution remains mathematically identical to the target model. Consequently, we focus exclusively on acceleration metrics rather than generation quality:

- **Wall-time Speedup:** The actual end-to-end speedup ratio relative to vanilla autoregressive decoding.

- **Mean Accepted Tokens (MAT):** The average number of tokens accepted per verification cycle. However, we argue that MAT is insufficient for evaluating dynamic drafting methods, as arbitrarily increasing the draft depth inevitably inflates the MAT without necessarily reflecting computational efficiency (Brown et al., 2024).

- **Draft Utilization ($u$):** To address the limitations of MAT, we introduce Draft Utilization, defined as the ratio of accepted tokens to the total number of drafted tokens per cycle ($u = \text{MAT}/\text{Depth}$). This metric provides a more precise measure of the drafting strategy's efficiency.

### 5.2. Main Results

**Low-Load Case (BS = 1).** Table 1 summarizes the performance of *ECHO* in the low-load regime. *ECHO* establishes a new SOTA speedup range of $1.63\times$–$5.35\times$ across all benchmarks, demonstrating three distinct advantages: **(1) Scalability to Industrial Models.** *ECHO* excels on massive architectures where speculation is challenging. On the industrial-grade **Qwen3-235B**, it attains a $\mathbf{2.02\times}$ average speedup, surpassing dynamic (DDD, $1.77\times$) and static (EAGLE-3, $1.69\times$) baselines by **14%** and **19%**, respectively. This validates the robustness of sparse gating against the sharp probability distributions characteristic of large-

*Table 1.* **Main results on five benchmarks under the low-load setting (BS = 1)**. Performance comparison between *ECHO* and existing baselines across diverse model configurations. Bold numbers denote the best speedup.

| Models | Methods | HumanEval | | GSM8K | | CNN/DM | | Alpaca | | MT-Bench | | Avg. |
|---|---|---|---|---|---|---|---|---|---|---|---|---|
| | | MAT | Speedup | MAT | Speedup | MAT | Speedup | MAT | Speedup | MAT | Speedup | |
| Vicuna-13B | Lookahead | 1.73 | 1.69× | 1.88 | 1.79× | 1.48 | 1.44× | 1.49 | 1.44× | 1.67 | 1.63× | 1.60× |
| | Sps | 2.55 | 1.81× | 1.99 | 1.75× | 2.31 | 1.71× | 2.01 | 1.74× | 2.25 | 1.81× | 1.76× |
| | Medusa | 2.78 | 2.25× | 2.63 | 2.12× | 2.09 | 1.65× | 2.44 | 1.96× | 2.58 | 2.08× | 2.01× |
| | Hydra | 3.87 | 2.75× | 3.66 | 2.60× | 2.82 | 1.95× | 3.51 | 2.48× | 3.64 | 2.53× | 2.46× |
| | OPT-Tree | 8.21 | 3.85× | 6.77 | 3.25× | 6.91 | 2.75× | 6.56 | 3.20× | 6.95 | 3.30× | 3.27× |
| | DDD | 8.95 | 4.95× | 6.21 | 3.92× | 6.02 | 3.45× | 5.98 | 3.59× | 6.05 | 3.95× | 3.97× |
| | EAGLE3 | 8.49 | 4.81× | 6.82 | 3.85× | 6.41 | 3.38× | 6.49 | 3.65× | 6.83 | 3.89× | 3.92× |
| | *ECHO* | 9.35 | **5.25×** | 6.53 | **4.08×** | 6.34 | **3.53×** | 6.24 | **3.74×** | 6.33 | **4.12×** | **4.14×** |
| LLaMA3.1-8B | DDD | 6.95 | 4.18× | 6.02 | 4.07× | 5.08 | 3.10× | 6.53 | 4.08× | 5.98 | 3.70× | 3.83× |
| | EAGLE3 | 7.18 | 4.02× | 6.50 | 3.94× | 5.46 | 3.02× | 7.06 | 4.01× | 6.43 | 3.63× | 3.72× |
| | *ECHO* | 7.22 | **4.30×** | 6.28 | **4.10×** | 5.24 | **3.26×** | 6.81 | **4.19×** | 6.23 | **3.86×** | **3.94×** |
| LLaMA3.3-70B | DDD | 6.82 | 5.13× | 6.15 | 4.63× | 4.92 | 3.52× | 6.68 | 4.84× | 5.70 | 4.02× | 4.43× |
| | EAGLE3 | 7.12 | 4.98× | 6.53 | 4.72× | 5.19 | 3.60× | 6.95 | 4.75× | 5.92 | 4.09× | 4.43× |
| | *ECHO* | 7.07 | **5.35×** | 6.41 | **5.08×** | 5.10 | **3.79×** | 6.84 | **4.94×** | 5.94 | **4.32×** | **4.70×** |
| Qwen3-8B | DDD | 3.65 | 2.54× | 3.72 | 2.32× | 3.08 | 2.13× | 3.22 | 2.06× | 3.48 | 2.35× | 2.28× |
| | EAGLE3 | 3.91 | 2.37× | 3.94 | 2.35× | 3.28 | 1.98× | 3.46 | 2.09× | 3.71 | 2.20× | 2.20× |
| | *ECHO* | 3.82 | **2.74×** | 3.88 | **2.68×** | 3.20 | **2.37×** | 3.36 | **2.51×** | 3.63 | **2.57×** | **2.57×** |
| Qwen3-32B | DDD | 2.82 | 2.18× | 3.15 | 2.38× | 2.38 | 1.71× | 2.68 | 2.11× | 2.88 | 1.97× | 2.07× |
| | EAGLE3 | 2.99 | 2.02× | 3.32 | 2.41× | 2.55 | 1.67× | 2.83 | 1.97× | 3.04 | 1.93× | 2.00× |
| | *ECHO* | 2.95 | **2.37×** | 3.29 | **2.72×** | 2.48 | **1.93×** | 2.74 | **2.31×** | 2.95 | **2.25×** | **2.32×** |
| Qwen3-235B | DDD | 2.20 | 1.88× | 2.48 | 1.68× | 2.05 | 1.49× | 2.18 | 1.78× | 2.28 | 1.99× | 1.77× |
| | EAGLE3 | 2.41 | 1.82× | 2.73 | 1.71× | 2.02 | 1.35× | 2.35 | 1.72× | 2.54 | 1.83× | 1.69× |
| | *ECHO* | 2.32 | **2.23×** | 2.59 | **1.92×** | 2.08 | **1.63×** | 2.24 | **2.07×** | 2.37 | **2.23×** | **2.02×** |

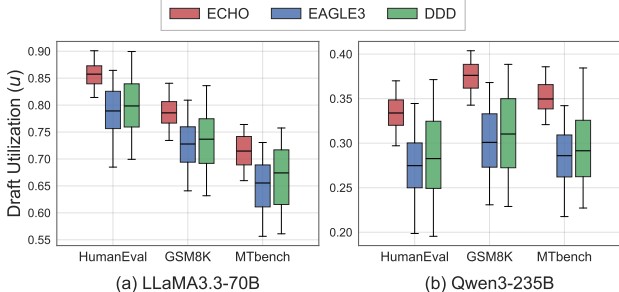

*Figure 4.* **Request-level Draft Utilization ($u$) across datasets and models.** We evaluate performance on LLaMA3.3-70B (a) and Qwen3-235B (b) across HumanEval, GSM8K, and MT-Bench. Each box shows the distribution of per-request Draft Utilization, spanning the 25th–75th percentiles, with whiskers covering the 5th–95th percentiles. *ECHO* consistently surpasses EAGLE-3 and DDD in average utilization.

scale models. **(2) Efficiency via Sparse Gating.** Unlike prior dynamic methods burdened by the overhead of dense, layer-wise evaluation, *ECHO* employs lightweight sparse gating to mitigate misjudgment accumulation. This efficiency allows *ECHO* to outperform the representative dynamic method (DDD) by **15.8%** on Qwen3-32B, effectively redirecting the saved computational budget toward effective token generation. **(3) High Draft Utilization and Robustness.** As illustrated in Figure 4, *ECHO* maximizes Draft Utilization ($u$) by dynamically aligning draft depth with

confidence. By proactively truncating uncertain branches, *ECHO* avoids the computational waste observed in static or overly aggressive drafting. Consequently, it achieves not only a higher average $u$ but also a significantly narrower interquartile range compared to baselines. This tighter distribution confirms *ECHO*'s superior robustness, ensuring consistent drafting precision across diverse inputs.

**High-Load Case (BS > 1).** We also evaluate the high-concurrency regime using SGLang to measure system throughput (tokens/s) across batch sizes from 8 to 256. Crucially, the efficacy of *ECHO* is governed by when the serving system enters the compute-bound verification regime. Smaller models remain memory-bound until high concurrency, whereas industrial-scale models saturate verification compute much earlier. Once saturation occurs, each iteration is effectively constrained by a fixed verification budget (Sec. 2), and any inefficiency in (i) which tokens enter the verification batch or (ii) how budget is allocated across heterogeneous requests directly translates into lost throughput. *ECHO* is a unified, budget-aware framework; however, different components dominate in different regimes. We analyze these two cases below.

**(1) Small-Scale Models: Late Compute-Bound and Precise Truncation.** For smaller models (e.g., LLaMA-3.1-8B and Qwen3-8B), the system is often not compute-bound until high concurrency. At low batch sizes, verifying ex-

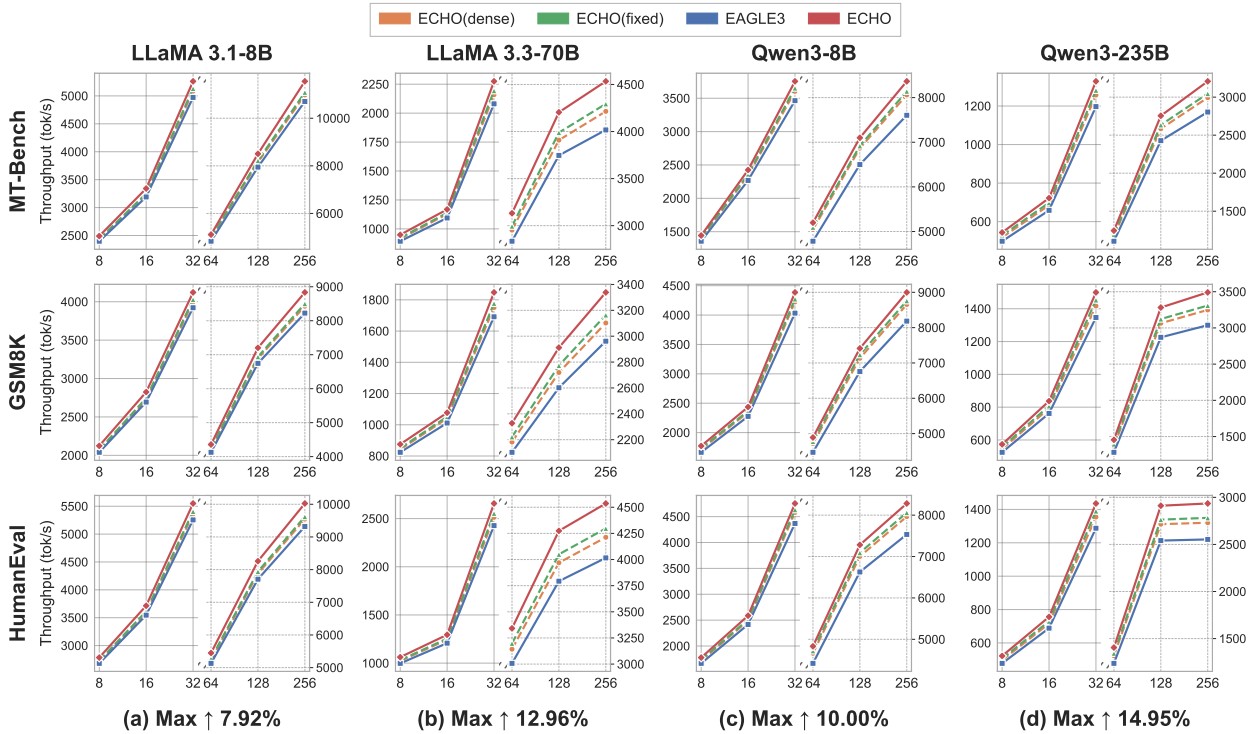

*Figure 5.* **Main results on High-Load Case (BS $> 1$).** We evaluate *ECHO* against EAGLE3 and two *ECHO* variants on three benchmarks using four model configurations. The maximum improvement percentage below each column is against EAGLE3.

tra speculative tokens hurts less because memory effects still dominate. Once verification becomes the bottleneck, EAGLE-3's fixed deep trees start wasting batch capacity on low-confidence branches. *ECHO* uses **sparse gating** to truncate these branches early, so fewer low-utility tokens enter verification. At BS $= 256$, this improves throughput by **8%** (10,703 $\rightarrow$ 11,551 tokens/s).

**(2) Industrial-Scale Models: Early Compute-Bound and Elastic Reallocation.** For Qwen3-235B, verification becomes compute-bound at much lower concurrency , so wasted verification work immediately reduces global throughput. Here, the main issue is not only bad branches within one request, but also how budget is shared across requests: EAGLE-3 spends too much on difficult requests while leaving little room to extend easy, high-confidence ones. *ECHO* applies **elastic budget scheduling** under the global cap (Eq. 4), shifting tokens saved from truncated low-confidence requests to deepen high-confidence requests. This yields a **14.4%** throughput gain at BS $= 256$ (2,803 $\rightarrow$ 3,207 tokens/s), showing that budget reallocation is key for compute-bound serving.

## 5.3. Ablation

To clarify the role of each component in *ECHO*, we study two design questions: where to apply gating (sparse vs. dense) and how to set gating thresholds (depth-aware vs. fixed). To understand which design choices matter, we

compare *ECHO* with two simplified variants: **Dense Gating** (making a gating decision for each depth) and **Fixed Threshold** (using a fixed threshold for all depths). Figure 5 shows that the full *ECHO* consistently performs best.

**Sparse Gating vs. Dense Gating.** Dense gating is a natural baseline: if we check more often, can we save more wasted tokens? The answer is no. On LLaMA-3.1-8B at BS $= 256$, **Dense Gating** is about **5%** worse than *ECHO* (11,551 $\rightarrow$ 10,978 tokens/s). This happens for two simple reasons. First, the checks themselves cost time; doing them at every step adds up, especially for small models. Second, confidence scores at many intermediate depths are not very reliable (Figure 2), so frequent checks can make wrong calls and cut branches that would have been accepted. By only checking at a few reliable "sweet spots", *ECHO* keeps the overhead low while making more accurate decisions.

**Depth-Aware Threshold vs. Fixed Threshold.** The **Fixed Threshold** baseline uses a single confidence cutoff $\tau$ for all depths. This is brittle because confidence naturally decreases with depth: deeper tokens often have lower probabilities even when they are correct. As a result, one threshold cannot work well everywhere: a high $\tau$ prunes too aggressively at deeper layers, while a low $\tau$ lets in too many low-quality tokens near the root. On Qwen3-235B, *ECHO* improves throughput over **Fixed Threshold** by **5.3%** (3,046 $\rightarrow$ 3,207 tokens/s), showing that depth-aware calibration of $\tau_d$ is important for using the verification budget effectively.

## 5.4. Robustness Across Workloads and Deployment Settings

Appendix D reports deployment-oriented checks covering calibration transfer, sampling temperature, long contexts, hardware portability, tail latency, and draft-side overhead. Thresholds calibrated on HumanEval transfer to GSM8K, CNN/DM, Alpaca, and MT-Bench with only 1–3% throughput loss relative to per-dataset calibration; across these datasets, calibrated thresholds differ by less than 0.05. The same thresholds also preserve *ECHO*'s advantage when temperature $> 0$, suggesting that sparse gating relies on relative confidence ranking rather than a probability scale tied to greedy decoding.

*ECHO* also remains effective under deployment stress tests. On PG19, it improves throughput over EAGLE-3 by more than 20% at BS=16 with 16K context; on H20 GPUs, the gain reaches 25%. We also discuss fairness and overhead. For fairness, truncation only removes low-confidence speculative tokens; each request still makes the same one-token progress guaranteed by standard autoregressive decoding. For overhead, our measurements show that target-model verification remains the dominant latency component in the evaluated EAGLE-style implementation, so the extra draft-side work does not become the main bottleneck.

## 6. Related Works

**Speculative Decoding** SD leverages parallel verification to accelerate inference (Leviathan et al., 2023; Chen et al., 2023). Existing approaches fall into three categories: draft model-based methods using separate lightweight drafters (Zhou et al., 2023); draft model-free methods utilizing auxiliary heads (e.g., Medusa (Cai et al., 2024), EAGLE (Li et al., 2025b)) or multi-token prediction (Zeng et al., 2025); and non-parametric methods relying on retrieval or matching logic (He et al., 2024). While these works optimize draft generation, maximizing efficiency further requires optimizing the structure of these candidates.

**Dynamic Token Tree** Tree-based SD explores multiple paths to maximize acceptance (Miao et al., 2024). Static tree methods employ fixed structures for low overhead but lack flexibility (Li et al., 2025b). Conversely, dynamic tree strategies (e.g., TALON (Liu et al., 2026), OPT-Tree (Wang et al., 2025)) adapt topology based on token probability. However, these methods typically rely on dense evaluation logic, introducing significant control overhead and generating irregular (ragged) batch shapes incompatible with standard high-performance serving kernels.

**SD in High-Concurrency Scenarios** Research has recently expanded from single-batch acceleration to system-level scheduling. While server-side (Fu et al., 2024) and client-side (Liu et al., 2024) optimizers exist, recent works like TETRIS (Wu et al., 2025) and TurboSpec (Liu et al., 2025a) have begun integrating SD into serving. These system-level schedulers are orthogonal and complementary to *ECHO*: they operate at the request or serving level, whereas *ECHO* focuses on intra-batch verification-budget allocation and dynamic tree construction. Direct comparison is limited by framework differences and, for TurboSpec, the lack of a public implementation. Nevertheless, the methods are conceptually composable: *ECHO* can be integrated with request-level schedulers to improve token-level allocation under strict verification limits.

## 7. Conclusion

We study SD under high-concurrency scenarios and show that verification becomes the bottleneck, invalidating "free-lunch" assumptions. We propose *ECHO*, a budget-aware framework that uses sparse gating and elastic scheduling to reduce verification waste while avoiding the error accumulation of fine-grained dynamic control. Experiments show consistent throughput gains across model scales, motivating a serving-oriented SD design that jointly optimizes global verification steps and per-step verification efficiency under compute-bound constraints.

## 8. Limitations

This work focuses on EAGLE-style trained drafters integrated into SGLang. The claim that draft-side overhead is negligible is therefore bounded to the architectures evaluated here; other drafting setups, including self-speculative decoding with early exits or layer skipping and two-model SD with smaller same-family draft models, can have different draft/verify ratios and require dedicated profiling. Lower-quality or poorly calibrated drafters may also change the reliability of confidence-based gates and should be tested separately. *ECHO* currently uses startup warm-up calibration; the observed cross-dataset stability is strong, but online threshold adaptation with sliding-window latency feedback remains future work for highly non-stationary production traffic. We evaluate contexts up to 16K tokens, leaving 32K–128K ultra-long-context serving as an open stress test.

## Impact Statement

This paper presents work whose goal is to advance the field of machine learning. There are many potential societal consequences of our work, none of which we feel must be specifically highlighted here.

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

# A. Theoretical Analysis

This appendix provides full derivations for the two formal results summarized in the main text (Theorem 1 and Theorem 2). We keep the exposition aligned with the serving-centric formulation in Sec. 2 and the fixed verification cap in Eq. 4.

## A.1. Coverage Gain via Width Expansion

*ECHO* triggers "Opportunistic Width Expansion" only when depth extension is halted due to low confidence and a residual budget exists. Here we formalize the probabilistic advantage of this structural shift.

As noted in Gumiho (Li et al., 2025a), the acceptance probability of a single path decays significantly as depth increases. At a truncation depth $d$ where the top-1 confidence is low, allocating the residual budget to width does not guarantee a longer accepted sequence immediately, but it **strictly increases the cumulative probability mass** that the ground-truth token is covered within the candidate set.

**Theorem 1** (Coverage Gain). *Let $\mathcal{S}_k = \{x^{(1)}, \ldots, x^{(k)}\}$ be the set of top-k candidate tokens at depth $d$, sorted by probability $p_t(x \mid x_{<d})$. Let $x^*$ be the ground-truth token sampled from $p_t$, and define the coverage probability as $\mathbb{P}(x^* \in \mathcal{S}) \triangleq \sum_{x \in \mathcal{S}} p_t(x)$. Expanding the candidate set size from $k$ to $k'$ (where $k' > k$) yields a strictly positive gain in the coverage probability:*

$$\mathbb{P}(x^* \in \mathcal{S}_{k'}) - \mathbb{P}(x^* \in \mathcal{S}_k) = \sum_{i=k+1}^{k'} p_t(x^{(i)} \mid x_{<d}) > 0, \tag{10}$$

*whenever $\sum_{i=k+1}^{k'} p_t(x^{(i)} \mid x_{<d}) > 0$.*

*Proof Sketch.* Since the target distribution has non-zero entropy (implied by the low confidence that triggered truncation), the probability mass is distributed across multiple tokens (i.e., $p_t(x^{(i)} \mid x_{<d}) > 0$ for some $i > k$). Equation 10 explicitly shows that widening the set accumulates these non-zero probabilities. This probabilistic gain minimizes the risk of the target model rejecting the entire node, thereby acting as a "safety net" when depth extension is too risky. $\square$

**Remark.** Theorem 1 formalizes why, once depth extension becomes unreliable, spending residual budget on width is a principled fallback: it improves coverage (probability of including $x^*$) even if it does not guarantee immediate depth progress.

## A.2. Batch-Level Objective Under Compute-Bound Constraints

This section supports the statement used in Theorem 2 (main text): under saturated compute-bound serving with a fixed verification cap, improving end-to-end throughput reduces to improving the batch-level expected accepted tokens per iteration.

**Setup.** Consider a single SD iteration with a batch of $B$ requests. Request $i$ submits $K_i$ draft candidates, and the target model accepts $L_i$ tokens (random variable). The total verification load is $K_{\text{total}} = \sum_{i=1}^{B} K_i$. In compute-bound serving (Sec. 2), verification latency grows approximately linearly with the verified token count (Eq. 2). *ECHO* enforces a per-iteration cap (Eq. 4) such that $K_{\text{total}} \leq K_{\max}$, and the system is typically operated near saturation, i.e., $K_{\text{total}} \approx K_{\max}$.

**Proposition 1** (Compute-Bound Objective Reduction). Under a fixed verification cap $K_{\text{total}} = K_{\max}$, the per-iteration latency is (approximately) constant. Consequently, maximizing end-to-end system throughput is equivalent to maximizing the **aggregate expected accepted length** per iteration:

$$\max \, \mathcal{J} \triangleq \max \sum_{i=1}^{B} \mathbb{E}[L_i]. \tag{11}$$

*Proof.* Throughput (accepted tokens per unit time) for an iteration is proportional to $\frac{\sum_{i=1}^{B} L_i}{T_{\text{iter}}}$. In the compute-bound regime, Eq. 2 implies $T_{\text{iter}}$ is dominated by verification cost and scales with $K_{\text{total}}$. If the scheduler enforces a fixed total verified token count $K_{\text{total}} = K_{\max}$ (Eq. 4), then $T_{\text{iter}}$ becomes (approximately) a constant for that iteration. Therefore, maximizing expected throughput reduces to maximizing the expected numerator, i.e., $\sum_i \mathbb{E}[L_i]$. $\square$

**Why this matters.** Proposition 1 implies that a policy can improve system performance even if it reduces the MAT of some requests, provided the batch-level aggregate $\sum_i \mathbb{E}[L_i]$ increases. This motivates elastic reallocation across requests in *ECHO*.

### A.3. Marginal Utility Exchange and Budget Reallocation

We now formalize when reallocating a small amount of budget from one request to another improves the batch objective in Eq. 11.

**Per-request response curve.** Let

$$f_i(k) \triangleq \mathbb{E}[L_i \mid K_i = k] \tag{12}$$

be the expected accepted length for request $i$ given $k$ verified candidates. Define the marginal gain of the $k$-th candidate token as

$$\Delta_i(k) \triangleq f_i(k) - f_i(k-1). \tag{13}$$

Intuitively, $\Delta_i(k)$ measures the incremental expected accepted tokens obtained by spending one additional verification token on request $i$.

**Theorem 2** (Marginal Utility Exchange)**.** *Consider two requests $i$ and $j$ sharing a fixed total budget $K$ within an iteration, with $\sum_{m=1}^{B} K_m = K_{\max}$. If the marginal gain of request $j$ at its next token exceeds the marginal gain of request $i$ at its current last token:*

$$\Delta_j(K_j + 1) > \Delta_i(K_i), \tag{14}$$

*then reallocating one token from $i$ to $j$ (i.e., $K_i \leftarrow K_i - 1$, $K_j \leftarrow K_j + 1$) strictly increases the batch objective $\mathcal{J} = \sum_{m=1}^{B} \mathbb{E}[L_m]$ and therefore improves end-to-end throughput under compute-bound serving (by Proposition 1).*

*Proof.* Let the original allocation be $(K_i, K_j)$ and the new allocation be $(K_i - 1, K_j + 1)$, with other $K_m$ unchanged. The change in the batch objective is

$$\begin{aligned} \Delta\mathcal{J} &= \big[f_j(K_j + 1) - f_j(K_j)\big] - \big[f_i(K_i) - f_i(K_i - 1)\big] \\ &= \Delta_j(K_j + 1) - \Delta_i(K_i). \end{aligned} \tag{15}$$

Under condition (14), $\Delta\mathcal{J} > 0$, hence the batch objective strictly increases. By Proposition 1, this strictly improves throughput in the compute-bound regime with a fixed verification cap. $\square$

**Implication for *ECHO*.** Sparse gating in *ECHO* is designed to identify low-confidence (hence low marginal-utility) continuations. When confidence is low, the marginal gain $\Delta_i(\cdot)$ diminishes due to probability decay along deeper paths; truncating such branches reduces spending on low-utility tokens. The freed budget can be (i) reallocated to other high-confidence requests for further depth extension, or (ii) used for width expansion at a truncation depth when no depth extension is available. Both behaviors align with Theorem 2: moving budget from lower $\Delta$ to higher $\Delta$ increases the batch-level objective, even if the MAT of some truncated requests becomes shorter.

# B. Extended Related Works

## B.1. Speculative Decoding

SD exploits the parallel verification capability of Transformers to accelerate autoregressive generation (Leviathan et al., 2023; Chen et al., 2023; Shen et al., 2025). The core paradigm involves a lightweight drafter proposing a sequence of tokens, which the target model verifies in a single forward pass. Draft model-based methods utilize a smaller, separate model (often quantized or distilled) to generate candidates (Zhou et al., 2023). While effective, they require maintaining two separate models and synchronizing their vocabularies. To mitigate this maintenance overhead, draft model-free methods have emerged. Approaches like Medusa (Cai et al., 2024) and EAGLE (Li et al., 2024a; 2025b) attach auxiliary prediction heads to the target model's frozen layers to predict future tokens. Recently, Multi-Token Prediction (MTP) (Zeng et al., 2025) co-trains these heads jointly with the main model. Additionally, non-parametric methods leverage retrieval or matching logic without additional training. Notable examples include prompt lookup decoding (Saxena, 2023), which reuses recurring phrases from the context, and retrieval-augmented SD (He et al., 2024; Shen et al., 2026), which fetches candidates from external corpora. While the aforementioned methods focus on the generation source of draft tokens, maximizing the verification efficiency further requires optimizing the structural organization of these candidates.

## B.2. Dynamic Token Tree

To further improve the acceptance rate per verification step, tree-based speculation explores multiple candidate paths simultaneously (Miao et al., 2024). Static tree methods rely on predetermined structures, where the tree shape (width and depth) is fixed manually or heuristically based on average acceptance rates (Cai et al., 2024; Li et al., 2025b). These methods prioritize high-probability tokens to form rigid layers but lack flexibility. Dynamic tree methods aim to adapt the tree structure to the current generation context. Approaches like TALON (Liu et al., 2026) and OPT-Tree (Wang et al., 2025) construct larger trees or dynamic widths based on token entropy and probability, applying adaptive pruning to fit within a compute budget. However, these methods typically employ dense, node-wise evaluation logic that incurs significant control overhead and often generates irregular (ragged) batch shapes incompatible with standard high-performance serving kernels.

## B.3. SD in High-Concurrency Scenarios

With the rise of production LLM serving systems, research has expanded from single-batch acceleration to system-level scheduling. Existing scheduling works can be broadly categorized into server-side approaches (Fu et al., 2024; Liu et al., 2023), which focus on maximizing overall throughput and device utilization, and client-side approaches (Liu et al., 2024), which optimize for user-perceived latency and fairness. Recent works like TETRIS (Wu et al., 2025) and TurboSpec (Liu et al., 2025a) have begun to integrate SD into serving systems, optimizing draft token selection to balance inference speed with profitability. These schedulers operate at the request or service level and are complementary to *ECHO*'s intra-batch budget allocation. In particular, ECHO's sparse gates can improve token-level allocation accuracy, while its packed ragged-tree execution can remove the uniform-length constraint that limits static batch-level speculation. Because TurboSpec is not publicly released and TETRIS uses a different serving stack, we avoid claiming a direct apples-to-apples comparison; an indirect BS=64 comparison shows *ECHO* achieving an 18.7% throughput gain over AR on LLaMA3.3-70B, compared with TETRIS's reported 12.3% gain on LLaMA3.1-70B. Most prior work treats the draft tree as a black box or ignores the compute-bound nature of verification in high-concurrency regimes. *ECHO* bridges this gap by proposing a serving-centric framework that jointly optimizes tree construction and budget scheduling under strict verification constraints.

## C. Evaluation Details

For reproducibility, we discuss the experimental setup (Section 5) in detail and the source code of this project will be made available at a later time.

### C.1. Data Configurations

In our experiments, we evaluate *ECHO* using the following dataset settings. To ensure comprehensive coverage, our benchmarks span code generation, mathematical reasoning, summarization, and general instruction following: HumanEval (Chen et al., 2021), GSM8K (Cobbe et al., 2021), CNN/DM (Nallapati et al., 2016), Alpaca (Taori et al., 2023), and MT-Bench (Zheng et al., 2023) following the set of EAGLE-3. The maximum generation length for these tasks is set to 1024 tokens.

### C.2. Detailed Baselines

**Baselines and Implementation**    To strictly evaluate the effectiveness of *ECHO* in production-grade serving, we benchmark it against a comprehensive set of competitive baselines, categorizing them into four distinct paradigms of speculative decoding:

- **Standard SD** (Chen et al., 2023; Leviathan et al., 2023): The fundamental *draft-then-verify* framework. We utilize the same draft models as *ECHO* but execute them sequentially without any tree-based parallel verification, serving as the baseline to measure the raw speedup gain over auto-regressive decoding.

- **Retrieval-based SD**: We compare against **Lookahead** (Fu et al., 2024), a representative method that accelerates inference solely using the target model. Lookahead employs Jacobi iteration to generate multi-branch candidates without requiring a separate draft model, providing a reference for architecture-agnostic acceleration.

- **Training-based Methods**: This category represents the current state-of-the-art. We include MLP-based approaches like **Medusa** (Cai et al., 2024) and **Hydra** (Ankner et al.), which attach lightweight decoding heads to predict multiple tokens in parallel. Crucially, we select **EAGLE-3** (Li et al., 2025b) as our primary static baseline. EAGLE-3 utilizes feature-level auto-regression with a multi-layer fusion mechanism and constructs a static draft tree with fixed depth and width. Comparing against EAGLE-3 allows us to directly demonstrate the advantages of *ECHO*'s elastic budget scheduling over rigid geometric constraints in high-concurrency regimes.

- **Dynamic Tree Methods**: To assess the efficacy of our sparse gating strategy, we compare against **OPT-Tree** (Wang et al., 2025) and **DDD** (Brown et al., 2024). These methods optimize the draft tree topology typically via dense, node-wise heuristics or "generate-then-prune" strategies. **Note on Fairness:** Since these methods were originally designed for EAGLE-2, we have re-implemented and adapted them to support the stronger EAGLE-3 backbone. This ensures that any performance gap is attributable to the tree scheduling policy (Dense Control vs. Sparse Gating) rather than the underlying draft model capability.

### C.3. Model Configurations

To validate performance, we select state-of-the-art open-source model pairs followed by EAGLE3 such as the Vicuna (yuhuili/EAGLE3-Vicuna1.3-13B, lmsys/vicuna-13b-v1.3), LLaMA3.1 (yuhuili/EAGLE3-LLaMA3.1-Instruct-8B, meta-llama/Llama-3.1-8B-Instruct), LLaMA3.3 (yuhuili/EAGLE3-LLaMA3.3-Instruct-70B, meta-llama/Llama-3.3-70B-Instruct), Qwen3-8B (AngelSlim/Qwen3-8B-eagle3, Qwen/Qwen3-8B), Qwen3-32B (AngelSlim/Qwen3-32B-eagle3, Qwen/Qwen3-32B) and Qwen3-235B (lmsys/Qwen3-235B-A22B-EAGLE3, Qwen/Qwen3-235B-A22B) for each task. All model weights are loaded in bfloat16 format for optimized GPU inference without quantization. As a training-free method, *ECHO* does not modify any draft model parameters during evaluation. We summarize the model configuration in Table 2.

### C.4. Evaluation Details

**Hardware and Framework.** All experiments are conducted on 8 NVIDIA H100 (80GB) GPUs. For latency-critical benchmarks ($BS = 1$), we utilize the HuggingFace `transformers` library. For high-concurrency scenarios evaluations ($BS > 1$), we integrate all methods into `SGLang`, an industrial-grade inference engine, to accurately measure throughput under realistic kernel constraints. We use greedy sampling (temperature $= 0$) for all methods to ensure deterministic reproducibility. Specifically, we test qwen235b in Figure.1 without *CUDA-Graph* and other experiments in high batch size

*Table 2.* Model configurations.

| Models | Layers | dim | FFN dim | Vocabulary size |
|---|---|---|---|---|
| Vicuna 68M | 2 | 768 | 3072 | 32000 |
| Vicuna 13B | 40 | 5120 | 13824 | 32000 |
| LLaMA-3.1 8B | 32 | 4096 | 14336 | 128256 |
| LLaMA-3.3 70B | 80 | 8192 | 28672 | 128256 |
| Qwen-3 8B | 36 | 4096 | 12288 | 151936 |
| Qwen-3 32B | 64 | 5120 | 25600 | 151936 |
| Qwen-3 235B | 94 | 4096 | 12288 | 151936 |

settings are all using the *CUDA-Graph*.

**Adaptive Calibration.** To ensure robustness and generalization across diverse model-dataset combinations, *ECHO* incorporates a lightweight warm-up phase prior to evaluation. During this phase, we analyze the layer-wise acceptance distribution of the draft model to adaptively identify discriminative "sweet spots" and calibrate gating thresholds. This mechanism allows *ECHO* to automatically tailor its sparse gating policy to the specific confidence landscape of each model. For instance, the calibrated thresholds $\tau_d$ at specific depths $d$ are set as follows: for LLaMA-3.1-8B, $\{d_0 : 0.2, d_5 : 0.35, d_8 : 0.5\}$; and for the larger Qwen3-235B, $\{d_0 : 0.15, d_3 : 0.3, d_5 : 0.5\}$.

**Configuration Protocols.** We adopt distinct configuration strategies for different concurrency regimes:

- **Low Concurrency** ($BS = 1$)**:** Both EAGLE-3 and *ECHO* employ the default configuration (Tree Depth=8, Top-$k$=10, Total Tokens=60) to maximize the theoretical upper bound of acceptance length.

- **High Concurrency** ($BS > 1$)**:** EAGLE3 is configured with their respective default parameters. In contrast, *ECHO* initializes with a streamlined configuration (Tree Depth=3, Top-$k$=3, Total Tokens=5) and activates its elastic scheduler. This allows *ECHO* to dynamically adjust the depth and width allocation per request based on the real-time batch size and global verification budget, ensuring optimal resource utilization in compute-bound regimes.

# D. Additional Experimental Results

### D.1. Calibration Robustness

We test whether thresholds calibrated on one dataset remain effective under distribution shift. Across HumanEval, GSM8K, CNN/DM, Alpaca, and MT-Bench, calibrated thresholds vary by less than 0.05. Reusing thresholds calibrated only on HumanEval degrades throughput by only 1–3% relative to per-dataset calibration, while still substantially outperforming EAGLE-3.

*Table 3.* Cross-dataset threshold robustness on Qwen3-8B. Fixed thresholds are calibrated on HumanEval and reused without retuning.

| Method | GSM8K | CNN/DM | Alpaca | MT-Bench |
|---|---|---|---|---|
| EAGLE-3 | 2.35× | 1.98× | 2.09× | 2.20× |
| *ECHO* (per-dataset) | 2.68× | 2.37× | 2.51× | 2.57× |
| *ECHO* (fixed thresholds) | 2.59× (-3%) | 2.31× (-2%) | 2.49× (-1%) | 2.48× (-3%) |

### D.2. Temperature and Sparse-Gate Granularity

Table 4 evaluates non-greedy decoding at temperature $= 1$ while reusing thresholds calibrated at temperature $= 0$. *ECHO*'s advantage persists, indicating that the sweet spots depend mainly on relative confidence ranking rather than absolute probabilities tied to greedy decoding.

*Table 4.* Results with temperature $= 1$ using thresholds calibrated at temperature $= 0$.

| Model | Method | HumanEval | GSM8K | CNN/DM | Alpaca | MT-Bench | Avg. |
|---|---|---|---|---|---|---|---|
| LLaMA3-8B | DDD | 3.51× | 3.41× | 2.53× | 3.14× | 3.05× | 3.13× |
| LLaMA3-8B | EAGLE-3 | 3.55× | 3.36× | 2.57× | 3.34× | 3.01× | 3.17× |
| LLaMA3-8B | *ECHO* | **3.73×** | **3.53×** | **2.71×** | **3.61×** | **3.18×** | **3.35×** |
| Qwen3-8B | DDD | 1.98× | 2.03× | 1.75× | 1.97× | 2.03× | 1.95× |
| Qwen3-8B | EAGLE-3 | 1.87× | 1.98× | 1.71× | 1.83× | 2.01× | 1.88× |
| Qwen3-8B | *ECHO* | **2.35×** | **2.27×** | **1.99×** | **2.21×** | **2.34×** | **2.23×** |

We also ablate a denser every-2-layer gating schedule. Sparse gating consistently performs better: frequent gates pass through high-overlap intermediate depths, where noisy confidence signals accumulate false decisions.

*Table 5.* Sparse gating vs. every-2-layer gating.

| Model | Method | HumanEval | GSM8K | CNN/DM | Alpaca | MT-Bench |
|---|---|---|---|---|---|---|
| LLaMA3-8B | *ECHO* (2-layer) | 4.21× | 4.01× | 3.12× | 4.11× | 3.73× |
| LLaMA3-8B | *ECHO* | **4.30×** | **4.10×** | **3.26×** | **4.19×** | **3.86×** |
| Qwen3-8B | *ECHO* (2-layer) | 2.57× | 2.51× | 2.31× | 2.39× | 2.46× |
| Qwen3-8B | *ECHO* | **2.74×** | **2.68×** | **2.37×** | **2.51×** | **2.57×** |

### D.3. Long Contexts and Cross-Hardware Portability

Table 6 reports PG19 results for Qwen3-8B across context lengths from 1K to 16K. At BS=16, the gain remains above 20% even with a 16K context, showing that *ECHO* applies beyond short-generation, high-concurrency settings.

We further evaluate Qwen3-8B on NVIDIA H20 GPUs. Because H20 has a lower compute-to-memory ratio than H100, it reaches the compute-bound regime earlier and amplifies the benefit of pruning wasted verification tokens.

For 200B-class deployments, modern quantized serving can sustain large effective batch sizes. On an 8×H20 node with 96GB per GPU, Qwen3-235B weights require approximately 235GB under W8 quantization. After reserving fixed buffers and activation workspace, C8 KV cache with GQA leaves enough memory for batch sizes exceeding 300 at a 16K context in

*Table 6.* Throughput gain of *ECHO* over EAGLE-3 on PG19 with Qwen3-8B (H100).

| BS / Context | 1K | 2K | 4K | 8K | 16K |
|---|---|---|---|---|---|
| 8 | +6% | +6% | +7% | +6% | +6% |
| 16 | +26% | +25% | +25% | +22% | +21% |

*Table 7.* Cross-hardware performance on H20 GPUs with Qwen3-8B.

| BS | *ECHO* tok/s | EAGLE-3 tok/s | Gain |
|---|---|---|---|
| 8 | 601.27 | 525.01 | +15% |
| 16 | 674.79 | 547.82 | +23% |
| 32 | 730.35 | 583.58 | +25% |

the profiled serving configuration. This capacity estimate is consistent with modern LLM serving practice, where continuous batching, KV-cache pooling, and prefill-decode disaggregation help maintain large effective batch sizes in production deployments.

### D.4. Tail Latency, Fairness, and Draft Overhead

*ECHO*'s priority rule does not starve low-confidence requests. When a request is truncated, it stops spending extra speculative budget on low-confidence branches, but it still receives AR-equivalent progress through target-model verification. The saved tokens reduce the shared verification workload and can lower per-step latency for the entire batch, including the truncated request. In contrast, a static tree may force the same hard request to wait while low-confidence speculative tokens from itself or other requests consume the full verification budget.

*Table 8.* Draft-to-verification latency ratios in the evaluated EAGLE-style setup.

| Model / setting | Draft:verify ratio | Interpretation |
|---|---|---|
| LLaMA3.3-70B, BS=64 | 1:15 | Verification dominated |
| Qwen3-235B, BS=64 | 1:32 | Verification dominated |

The draft-side overhead claim is scoped to the evaluated EAGLE-style setup. Other drafting setups, including self-speculative early-exit or layer-skipping methods and two-model SD with smaller same-family drafters, can have different drafting costs. We leave a full overhead analysis for those settings to future work.

# E. Layer-wise Confidence Density Visualization for Each Model

## E.1. Visualization of Layer-wise Confidence Shifts

We provide a comprehensive visualization of layer-wise confidence distributions to characterize the stochastic nature of drafting across varying architectures. As illustrated in Figures 6 through 9 , we profile the probability density of Accepted (Red) and Rejected (Blue) tokens up to **Depth=8** for the LLaMA series and **Depth=5** for the Qwen series.

**The Stochastic Drift.**    These visualizations serve as the empirical foundation for *ECHO*. A critical oversight in prior dynamic methods is the assumption of a static confidence threshold. The plots reveal a distinct **Entropy Drift**: at shallow depths (e.g., Depth 0-1), the distributions are bimodal with a clear decision boundary, accepted tokens cluster near probability $0.9$, while rejected tokens cluster near $0.1$. However, as the tree deepens, the confidence of valid tokens naturally decays, causing the red distribution to migrate leftward and merge with the blue distribution. This creates the **"Overlap Regions"** (visualized in purple/grey), high-entropy zones where binary classification becomes inherently ambiguous. Operating blindly in these regions leads to error accumulation, manifesting as either the premature pruning of valid paths (false negatives) or the wasteful extension of invalid ones (false positives).

**Analysis of Distributional Dynamics.**    The density landscapes reveal distinct behaviors governed by model architecture and alignment:

- **Depth-Dependent Decay:** A universal trend is observed where the probability mass of the optimal path migrates from the high-confidence "Accepted" cluster toward the "Rejected" cluster as depth increases, blurring the separability.

- **Alignment Sensitivity:** The velocity of this drift correlates with the intrinsic draft-target alignment. Models with weaker alignment exhibit a more rapid transition into the ambiguous overlap zone at shallower depths.

**Operationalizing the "Sweet Spot".**    This variance substantiates the design of our **Sparse Gating**. As explicitly annotated by the **Green Dashed Lines** in the figures, *ECHO* identifies **"Sweet Spots"** layers where the overlap is minimal and the signal-to-noise ratio is maximized. Instead of applying a uniform threshold, *ECHO* utilizes a warm-up phase to profile these specific distributional shifts. By adaptively triggering verification gates only at these identified "Sweet Spots" and dynamically calibrating thresholds ($\tau_d$) to the local distribution, *ECHO* circumvents the high-entropy overlap zones, ensuring robust generalization across diverse model-dataset combinations.

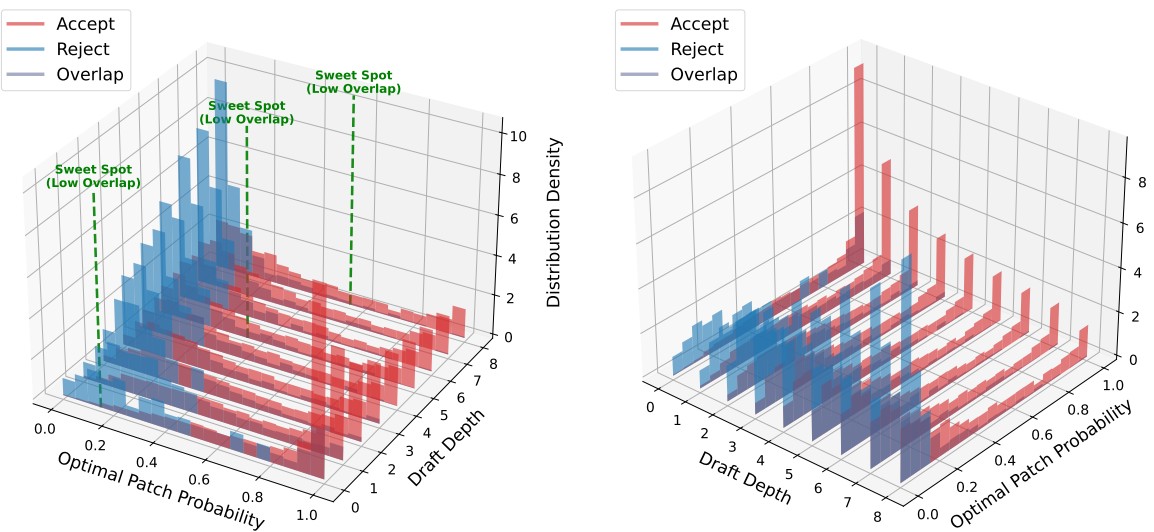

*Figure 6.* Visualization of confidence distributions across draft depths (LLaMA3.1-8B on MT-Bench).

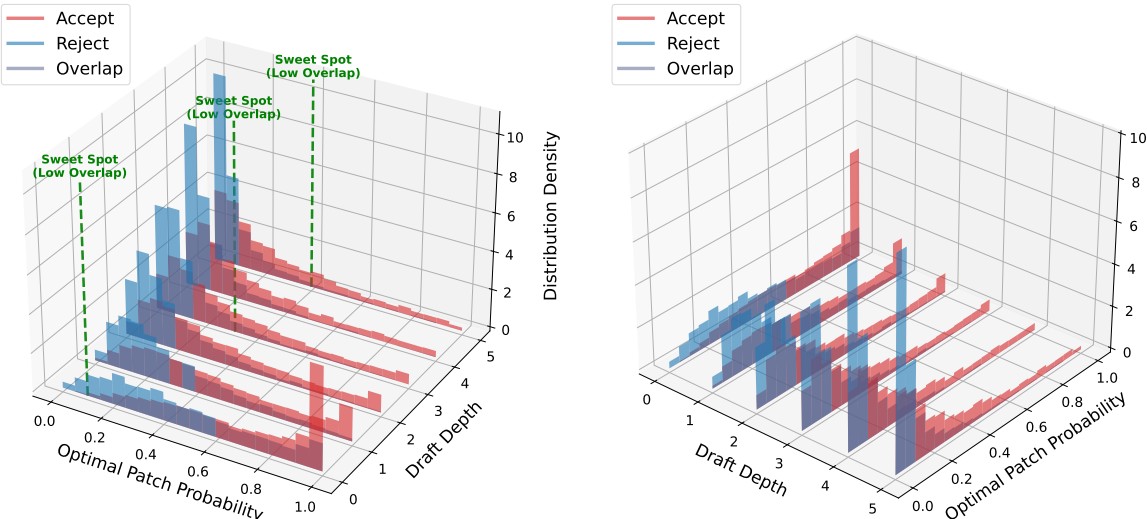

*Figure 7.* Visualization of confidence distributions across draft depths (Qwen3-8B on MT-Bench).

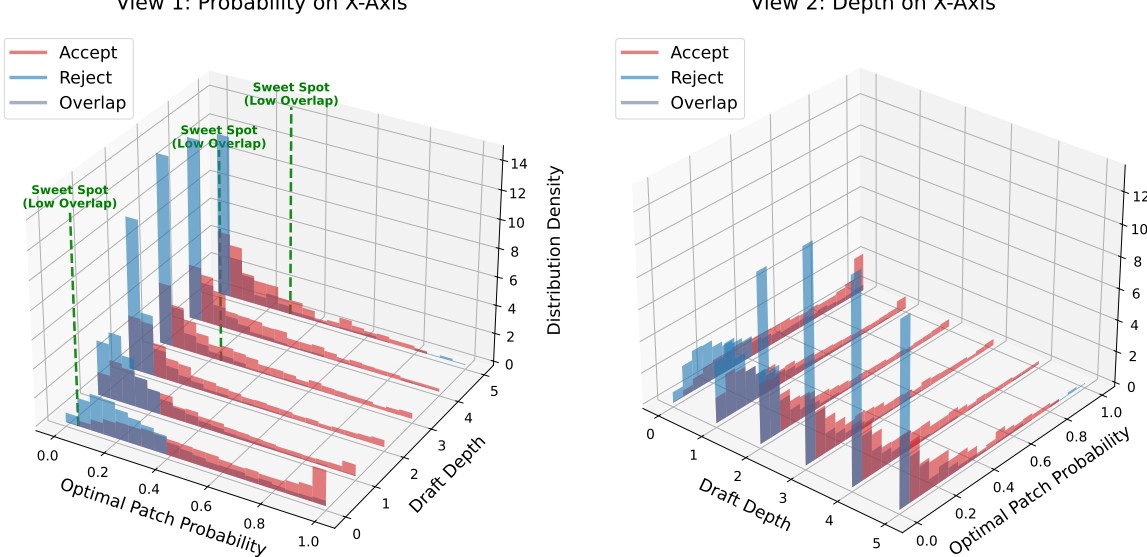

*Figure 8.* Visualization of confidence distributions across draft depths (Qwen3-32B on MT-Bench).

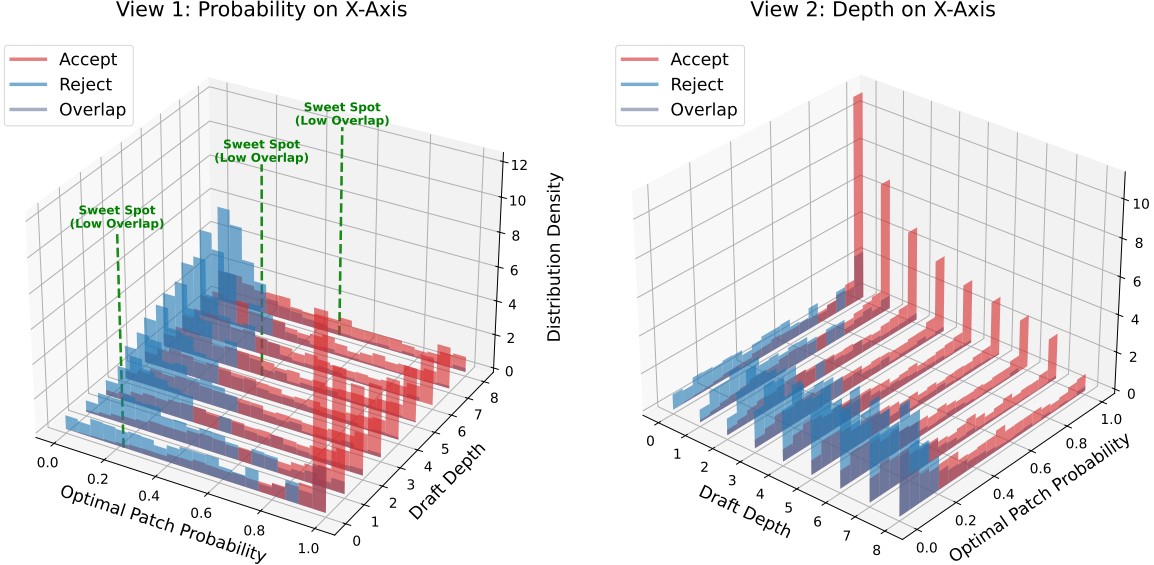

*Figure 9.* Visualization of confidence distributions across draft depths (LLaMA3.3-70B on MT-Bench).

