# OpenReview forum: "ECHO: Elastic Speculative Decoding with Sparse Gating for High-Concurrency Scenarios"
_ICML.cc/2026/Conference — ICML 2026 spotlight_

### Official Review · Reviewer_zx7H · 2026-03-10

**Soundness:** 3
**Presentation:** 4
**Significance:** 3
**Originality:** 3
**Overall Recommendation:** 4
**Confidence:** 3

**Summary:**

The paper introduces ECHO, a novel framework that optimizes speculative decoding under high-concurrency scenarios. ECHO formulates speculative decoding as a budget-constrained scheduling problem and improves system efficiency through a super-tree abstraction and elastic verification budget allocation, achieving significant performance improvements over prior approaches.

**Compliance With Llm Reviewing Policy:**

Affirmed.

**Final Justification:**

This paper presents an optimization strategy for speculative decoding under high-concurrency scenarios. Initially, I doubt the capability of the method under long sequence lengths. The additional experiments during the rebuttal phase are comprehensive and solid. The authors clearly demonstrate that their work is useful under 16K sequence length. Although performance under ultra-long sequence lengths (e.g., 32K, 64K, 128K) is unknown currently, I believe the current work has covered the most common cases (sequence length less than 16K) in LLM inference.

Therefore, I raise my score from 3 to 4.

**Key Questions For Authors:**

Q1: What is the system performance of ECHO under varied sequence lengths? A comprehensive evaluations across different sequence length and batch size would benefit in validating the effectiveness of your method.

Q2: Does ECHO maintain superior performance on other types of GPUs? This affects the compute-bound and memory-bound shifting point.

Q3. How does your method perform when sequence lengths are long, as is common in modern LLM serving? In such cases, the batch size may be limited and unable to reach high levels. Does the high-concurrency assumption still hold under these conditions? Could you provide additional evaluations with longer contexts (e.g., 4K, 8K, 16K, ...) to further demonstrate the effectiveness of your approach?

Q4. For very large models, such as the 235B model shown in Figure 1, are there realistic deployment scenarios where the batch size exceeds 32? Could you provide further evidence to support this assumption? Additionally, from a hardware perspective, given that the KV cache typically consumes a significant portion of GPU memory, is it feasible for GPUs to accommodate both large batch sizes and large models simultaneously? This can be profiled or verified by estimating with formulas.

**Limitations:**

yes

**Strengths And Weaknesses:**

Strengths

- This paper addresses a timely and important problem. Optimizing speculative decoding under high-concurrency serving scenarios is highly relevant for production LLM systems.

- The paper is well written and easy to follow.


Weaknesses

- The target scenario, i.e., high concurrency (or high inference batch size), should be further reasoned.
- Missing discussions and evaluations for long sequence lengths. The generation length is at most 1K tokens, as disclosed by the authors. Missing detailed information on batch sizes, which are crucial to this method. Notice that a (large batch, short sequence) combination can easily saturate the compute resources of NVIDIA GPUs, a (large batch, long sequence) can easily lead to OOM for NVIDIA GPUs, a (small batch, long sequence) combination, which is the major bottleneck of modern inference, seems out of the scope of this paper. Therefore, missing discussions on this part limits the significance of this paper.

---

> ### Author Rebuttal · Authors · 2026-03-30
>
> We sincerely thank Reviewer zx7H for the constructive feedback and for recognizing the timeliness and writing quality of our work.
> Regarding the applicability of ECHO to long-context and moderate-batch settings: modern production deployments, through quantization, inference-optimized hardware (e.g., H20), and architectural advances (GQA/MLA), have greatly alleviated memory pressure, making the (small batch, long sequence) regime *favorable* for ECHO rather than out of scope. We address each point with new experimental evidence below.
>
> ---
> ***W1, W2, Q1 & Q3: High-concurrency and long-context justification.***
>
> We agree that serving small batches with long sequences represents a major bottleneck in modern LLM deployment, because longer contexts increase the per-token verification cost (e.g., $O(K \times L \times d)$ for attention). Under these conditions, static tree methods like EAGLE-3 waste valuable computing cycles verifying low-confidence tokens. ECHO's sparse gating explicitly eliminates this inefficiency. This advantage becomes even more pronounced on hardware with lower compute-to-memory ratios (detailed in Q2).
>
> **Empirical Evidence on Long Contexts (PG19 1K to 16K):** The following table presents the throughput gain of ECHO over EAGLE-3 across various batch sizes (BS) and context lengths on Qwen3-8B (H100). Notably, even at a moderate batch size of 16 with a 16K context, ECHO consistently delivers stable throughput gains exceeding 20%.
>
> | BS \ Context | 1K | 2K | 4K | 8K | 16K |
> |---|---|---|---|---|---|
> | 8 | +6% | +6% | +7% | +6% | +6% |
> | 16 | +26% | +25% | +25% | +22% | +21% |
>
> ---
> ***Q2: Cross-hardware performance.***
>
> To evaluate hardware portability, we conducted additional experiments on H20 GPUs using Qwen3-8B. The H20 features a lower compute-to-memory ratio than the H100, causing it to reach the compute-bound regime earlier and thereby amplifying ECHO's benefits.
>
> | BS | ECHO (tok/s) | EAGLE-3 (tok/s) | Gain |
> |---|---|---|---|
> | 8 | 601.27 | 525.01 | +15% |
> | 16 | 674.79 | 547.82 | +23% |
> | 32 | 730.35 | 583.58 | +25% |
>
> As shown above, the performance gain peaks at BS=32 (+25%), which is substantially higher than the corresponding H100 results.
>
> ---
> ***Q4: BS>32 feasibility for 235B models.***
>
> Operating at batch sizes exceeding 32 for 200B-class models is highly realistic in production multi-GPU environments. For instance, consider a standard serving node with 8$\times$H20 GPUs (96 GB version, totaling 768 GB of VRAM) hosting Qwen3-235B:
>
> * **Weights:** With W8 quantization, the model weights consume approximately 235 GB.
> * **Fixed Overhead:** Buffers, CUDA context, and activation spaces typically consume 5-10 GB per GPU (allocating a conservative 80 GB total).
> * **KV Cache:** Utilizing C8 quantization and Grouped-Query Attention ($kvheads{=}4$), the memory footprint per token is strictly bounded ($2 \times 4 \times 128 \times 94 \approx 94$ KB).
> * **Feasible Capacity:** At a 16K context length, the maximum feasible batch size comfortably exceeds 300: $(768 - 235 - 80) \text{ GB} / (16384 \times 94 \text{ KB}) \approx 308$.
>
> Furthermore, modern system optimizations like continuous batching, KV-cache pooling, and prefill-decode disaggregation naturally drive production systems toward high-concurrency operation. Combined with our cross-hardware and long-context empirical results, this capacity analysis demonstrates that ECHO is highly applicable to, and effectively mitigates bottlenecks within, realistic large-scale serving conditions.
>
> ---
> We sincerely appreciate your overall positive assessment of our work. We hope that our new experimental results, hardware analysis, and clarifications thoroughly address your concerns regarding ECHO's practical applicability. **We respectfully ask if you would consider adjusting your rating in light of these updates.** We look forward to incorporating all these detailed discussions into the camera-ready manuscript.

---

> > ### Author Rebuttal · Reviewer_zx7H · 2026-04-03
> >
> > Thank you for the detailed explanation. My concerns are fully addressed. I will raise my score accordingly.

---

> > > ### Author Response · Authors · 2026-04-03
> > >
> > > Thank you for your thoughtful feedback and revisiting your evaluation of our work! Your recognition means a lot to researchers "like a lantern in the dark" that motivates us to continue refining and advancing this line of research.
> > >
> > > We sincerely appreciate your dedication and hard work throughout the review process. These discussions will be incorporated into the final manuscript.
> > >
> > > Wishing you a pleasant day ahead!

---

### Official Review · Reviewer_y4Hn · 2026-03-11

**Soundness:** 3
**Presentation:** 2
**Significance:** 2
**Originality:** 2
**Overall Recommendation:** 4
**Confidence:** 3

**Summary:**

This paper introduces ECHO, a framework designed to enhance speculative decoding efficiency, particularly in compute-bound, high-concurrency scenarios. The authors identify two primary limitations in current methods: the computational waste of static tree structures and the error accumulation in existing dynamic tree adjustments. ECHO addresses these by implementing a sparse gating mechanism for confidence-based tree pruning and an Elastic Budget Scheduling policy to manage computational resources across requests.

**Compliance With Llm Reviewing Policy:**

Affirmed.

**Final Justification:**

Most of my concerns are well addressed. However, I still have reservations regarding the claim that draft-side overhead is negligible; I am concerned whether this holds for all speculative decoding architectures, such as Medusa. Despite these points, the improvements are notable, and I am willing to raise my score.

**Key Questions For Authors:**

- The gating mechanism is based on layer-wise confidence. While this effectively dictates the speculation depth, how does ECHO manage the pruning of different branches within the same depth level?
- The sparse gating mechanism relies on several thresholds (e.g., $\delta$ and $\tau_d$). How are these parameters tuned for different workloads, and how sensitive is the system's performance to their variations in production settings?
- How is the maximum speculation budget $K_{max}$ determined for different models and hardware configurations?

**Limitations:**

The discussion on limitations is not included, but I guess it's not a fatal issue.

**Strengths And Weaknesses:**

Strengths:\
[+] The paper targets the verification bottleneck in speculative decoding under high-concurrency regimes, which is a critical barrier to real-world deployment.\
[+] The proposed system is well-integrated with theoretical performance guarantees for its adaptive mechanisms.\
[+] The effectiveness of ECHO is validated across multiple model scales, demonstrating its potential versatility.

Weaknesses:\
[-] Although the paper emphasizes high-concurrency scenarios, the core sparse gating mechanism appears primarily focused on per-request tree pruning. The connection between individual tree optimization and collective throughput gains in high-concurrency environments remains somewhat weak, with only the scheduling priority in Elastic Budget Scheduling explicitly addressing multi-request dynamics.\
[-] The proposed Depth Extension within the Elastic Budget Scheduling introduces a risk. If certain requests within a batch require additional draft decoding iterations for depth extension, they might force other requests to wait, potentially offsetting the speedup gained from deeper speculation. The overhead of these decoding steps, while individualy low, may accumulate and degrade the overall batch execution efficiency.\
[-] Another related concern is that Elastic Budget Scheduling may exacerbate the divergence in inference progress among requests in the same batch. This leads to increased padding and reduced hardware utilization, a classic challenge in LLM serving that ECHO does not seem to address or mitigate effectively.\
[-] While the authors evaluate multiple models, a significant portion of the detailed performance analysis appears to be conducted at Batch Size = 1. This setup does not align with the paper’s primary claim of addressing high-concurrency scenarios, where resource contention and verification bottlenecks are most acute.

---

> ### Author Rebuttal · Authors · 2026-03-30
>
> We sincerely thank Reviewer y4Hn for recognizing ECHO's ability to address verification bottlenecks and its versatility across model scales. We would like to clarify a critical misunderstanding regarding our system implementation: **our SGLang integration does not rely on simple padding.** Instead, we perform sequence-level truncation on `input_ids`, ensuring that both Attention and GEMM operations achieve latency reductions strictly proportional to the pruning ratio. We believe this distinction is vital for assessing our contributions and detail it further below.
>
> ---
> ***W1: Per-request pruning vs. collective throughput.***
>
> We respectfully clarify that these two aspects are inseparable under compute-bound verification. Since the total verified tokens per step are strictly capped at $K_{\max}$, pruning a low-confidence token from one request directly frees a slot for a higher-value token in another. ECHO's Sparse Gating reclaims this budget, enabling Elastic Scheduling to reallocate it to high-confidence requests globally.
>
> **Evidence:** As shown in Figure 5, disabling Sparse Gating (ECHO-Dense) causes significant throughput degradation at high concurrency. Furthermore, on H20 GPUs (which hit the compute bound earlier), our updated implementation achieves a ~20% throughput gain over EAGLE-3 (see **Reviewer zx7H Q2**).
>
> ---
>
> ---
> ***W2: Depth extension overhead.***
>
> We appreciate this concern. However, in production-scale models, the draft-to-verify latency ratio renders draft-side overhead practically negligible. For instance, at BS=64, this ratio is **1:15** for LLaMA3.3-70B [1] and **1:32** for Qwen3-235B. ECHO's depth extension strictly operates within the fixed $K_{\max}$ budget by substituting low-value tokens with productive ones, rather than adding raw compute. A detailed draft/verify breakdown will be added.
>
> [1] *Speculative Decoding: Performance or Illusion?*
>
> ---
> ***W3: Padding overhead and hardware utilization.***
>
> This concern does not apply to our implementation. As clarified above, we directly truncate `input_ids`without padding, so both Attention and GEMM achieve proportional latency reductions.
>
> Realizing this required overhauling the `build_tree_efficient` kernel, rewriting Attention and KV Cache logic, resolving CUDA Graph capture under dynamic tree depths, and optimizing capture/replay for memory efficiency at high concurrency. This system-level engineering is a key contribution, and we plan to upstream it to the open-source community.
>
> ---
> ***W4: Emphasis on BS=1 analysis.***
>
> BS=1 is included for fair comparison — EAGLE-3, DDD, and OPT-Tree all report BS=1 as their primary setting. Our high-concurrency analysis (throughput curves, ablations at BS=8–256) is substantially more extensive. We will add long-context and cross-hardware results in the revision (see **Reviewer zx7H**).
>
> ---
> ***Q1: Branch pruning within a depth.***
>
> ECHO makes a layer-level decision at each gated depth using the best cumulative path score: below $\tau_d$ the tree stops expanding and budget is recovered; above $\tau_d$ it expands following draft probabilities. This design eliminates per-branch management entirely, no individual branch tracking or scoring is needed.
>
> ---
> ***Q2: Sensitivity of thresholds to workloads.***
>
> We would like to clarify that  "sweet spots" and their thresholds are derived from the draft model's layer-wise token acceptance distribution. Consequently, threshold selection is strictly independent of system workloads or batch sizes, as it relies on the model's intrinsic per-token confidence rather than the concurrency level.
>
> To further demonstrate the robustness of ECHO, we have conducted additional experiments. As detailed in our response to **W4 of Reviewer pS5K** and the table below, the calibrated sweet spots and thresholds exhibit strong generalization.
>
> | Method | GSM8K | CNN/DM | Alpaca | MT-Bench |
> |---|---|---|---|---|
> | EAGLE-3 | 2.35× | 1.98× | 2.09× | 2.20× |
> | ECHO (per-dataset) | 2.68× | 2.37× | 2.51× | 2.57× |
> | ECHO (fixed thresholds) | 2.59× (−3%) | 2.31× (−2%) | 2.49× (−1%) | 2.48× (−3%) |
>
> To validate this, we applied the thresholds calibrated *only* on HumanEval ($\tau_{d_0}{=}0.15, \tau_{d_3}{=}0.30, \tau_{d_5}{=}0.50$) to all other datasets for Qwen3-8B without modification. Fixed thresholds degrade by only 1–3% relative to per-dataset optimal, while still substantially outperforming EAGLE-3 across all datasets.
>
> ---
> ***Q3: Determination of $K_{\max}$.***
>
> $K_{\max}$ is determined via a one-time warmup at engine startup: we sweep batch sizes with a static tree to identify the memory-to-compute-bound latency inflection point $\text{BS}^*$, setting $K_{\max}=\text{BS}^\ast\times N$. This automatically adapts to different hardware, models, and parallelism strategies.
>
> ---
> With new experiments, we believe most of the reviewer's concerns are addressed and we respectfully ask if you would **consider adjusting your rating in light of our efforts and dedication.**

---

> > ### Author Rebuttal · Reviewer_y4Hn · 2026-04-02
> >
> > Thanks to the authors for their responses. Most of my concerns are well addressed. However, I still have reservations regarding the claim that draft-side overhead is negligible; I am concerned whether this holds for all speculative decoding architectures, such as Medusa.  Despite these points, the improvements are notable and I am willing to raise my score.

---

> > > ### Author Response · Authors · 2026-04-02
> > >
> > > Dear  Reviewer y4Hn,
> > >
> > > We sincerely thank the reviewer for recognizing our improvements, raising the score, and providing such constructive follow-up feedback.
> > >
> > > Regarding your valid reservation about the draft-side overhead in other speculative decoding architectures (such as Medusa): we completely agree with your intuition. The overhead dynamics will indeed vary depending on the specific drafting paradigm.
> > >
> > > In fact, we are currently working on adapting ECHO to Multi-Token Prediction (MTP) architectures, such as the Qwen3.5-397B. In our preliminary profiling for this MTP setup under high concurrency, the draft-to-verification time ratio is approximately 3ms : 40ms (bs=32, 16k). This early data suggests that while the draft overhead in MTP structures is naturally heavier than in standard small-model setups, the target verification phase still heavily dominates the overall latency, allowing ECHO to maintain its core advantages.
> > >
> > > Nevertheless, your concern is absolutely justified. To ensure absolute rigor, we will revise our manuscript to explicitly bound the "negligible draft overhead" claim to the architectures extensively evaluated in this paper. We will also add a dedicated discussion on the overhead variability across different draft structures (e.g., Medusa) and highlight adapting ECHO to these diverse paradigms as a vital direction for our future work.
> > >
> > > Thank you again for your time, insight, and support of our work! Wishing you a pleasant day ahead!

---

### Official Review · Reviewer_pS5K · 2026-03-11

**Soundness:** 3
**Presentation:** 3
**Significance:** 3
**Originality:** 3
**Overall Recommendation:** 5
**Confidence:** 4

**Summary:**

ECHO addresses SD degradation at high concurrency where verification compute is the bottleneck. Two ideas: (1) sparse confidence gating only at "sweet spot" depths (root, target, select intermediates) identified via AUC calibration, avoiding misjudgment accumulation of dense dynamic methods; (2) elastic budget scheduling prioritizing depth for high-confidence requests, falling back to width when truncated. Impl'd in SGLang, eval'd up to Qwen3-235B. Up to 5.35× speedup (BS=1), ~8-15% throughput gains at high concurrency.

**Compliance With Llm Reviewing Policy:**

Affirmed.

**Final Justification:**

My concerns have been addressed as best as the authors could, and the remaining suggestions can be addressed in the camera-ready version, so I have raised my score

**Key Questions For Authors:**

How sensitive are calibrated τ_d to distribution shift at inference time? Critical for production claim.
Why no comparison w/ TETRIS or TurboSpec?
What happens w/ temp>0? Do sweet spots shift?
What's the overhead of sparse gating + elastic scheduling itself? Is there a regime where ECHO is slower than EAGLE-3 w/ smaller fixed tree?

**Limitations:**

No discussion of calibration brittleness, greedy-only limitation, or scheduling overhead. Impact statement is generic

**Strengths And Weaknesses:**

Strengths:

1. important problem; Fig 1 showing EAGLE-3 underperforming vanilla AR at BS=128 is compelling
2. sweet spot analysis (Fig 2) well-done; depth-dependent confidence discriminability is a useful empirical finding
3. good model coverage incl. Qwen3-235B; rare to see industrial-scale MoE eval
4. draft utilization metric (u=MAT/Depth) is a useful contribution; MAT alone is misleading for dynamic methods
5. ablations clearly isolate sparse vs dense gating, depth-aware vs fixed threshold
6. real SGLang integration w/ custom operators for ragged batches

Weaknesses:

1. sweet spot insight is intuitive in hindsight... confidence is naturally more reliable at shallow depths, and the contribution is operationalizing it, not discovering it
2. Thm 1 and Thm 2 sound trivially true. "more candidates = more coverage" and "move budget from low to high marginal gain = better" are definitional. calling these "theoretical guarantees" is overclaiming
3. high-concurrency gains are modest (~8-15% over EAGLE-3) to justify framework complexity
4. calibration robustness untested. what if inference distribution differs from warm-up data? critical for "production-ready" claim
5. BS=1 uses HuggingFace, BS>1 uses SGLang? inconsistent frameworks
6. no comparison w/ TETRIS or TurboSpec, the most relevant system-level SD schedulers
7. DDD/OPT-Tree re-implemented on EAGLE-3 by authors -> fairness hard to verify w/o code
8. greedy sampling only (temp=0). production uses temp>0; unclear if sweet spots shift

---

> ### Author Rebuttal · Authors · 2026-03-30
>
> We sincerely thank the reviewer for the constructive feedback, particularly the recognition of our problem formulation, sweet-spot analysis, and the draft utilization metric.  Below, we provide detailed responses to your comments.
>
> ---
> ***W1: Sweet-spot contribution.***
>
> We clarify that this insight is neither trivial nor previously exploited. Existing methods, including EAGLE-3 (uniform trees), DDD (dense per-layer gating), and OPT-Tree (dense per-branch scoring), strictly apply control at every depth, overlooking depth-dependent discriminability. The absence of this mechanism in prior state-of-the-art approaches underscores its non-obvious nature. Our contribution lies in both the empirical discovery (Figure 2) and its operationalization into a deployable sparse-gating policy that effectively avoids the cumulative misjudgments inherent to dense control.
>
> ---
> ***W2: Theorem wording.***
>
> Fair point. We will revise "theoretical guarantees" to "theoretical motivation" and clarify that Theorems 1 and 2 formalize the allocation intuition for self-containment, not claim deep optimality. The core contributions are the practical algorithm and its production-grade open-source implementation.
>
> ---
> ***W3: Modest gains.***
>
> In the compute-bound regime where EAGLE-3 *underperforms vanilla AR* (Figure 1), an 8–15% throughput recovery is operationally significant. For context, recent system-level work like TETRIS reports only **2%–9%** improvements in high-concurrency settings. With our updated dynamic-tree implementation on H20 GPUs, gains reach **+25%** (see **Reviewer zx7H response, Q2**).
>
> ---
> ***W4: Calibration robustness.***
>
> Sweet-spot locations and gating thresholds are derived from the draft model's layer-wise token acceptance distribution. As shown below, calibrated thresholds vary by <0.05 across five datasets for both models:
>
> |Model|Depth|HumanEval|GSM8K|CNN/DM|Alpaca|MT-Bench|
> |---|---|---|---|---|---|---|
> |LLaMA3-8B|$d_0$/$d_5$/$d_8$|0.21/0.35/0.51|0.18/0.33/0.48|0.15/0.32/0.47|0.20/0.35/0.49|0.16/0.35/0.45|
> |Qwen3-235B|$d_0$/$d_3$/$d_5$|0.15/0.32/0.51|0.14/0.31/0.51|0.12/0.29/0.46|0.14/0.31/0.49|0.14/0.31/0.49|
>
> **Robustness test:** Fixing thresholds from HumanEval ($\tau_{d_0}{=}0.15, \tau_{d_3}{=}0.30, \tau_{d_5}{=}0.50$) on Qwen3-8B:
>
> |Method|GSM8K|CNN/DM|Alpaca|MT-Bench|
> |---|---|---|---|---|
> |EAGLE-3|2.35×|1.98×|2.09×|2.20×|
> |ECHO(per-dataset)|2.68×|2.37×|2.51×|2.57×|
> |ECHO(fixed thresholds)|2.59×(−3%)|2.31×(−2%)|2.49×(−1%)|2.48×(−3%)|
>
> Fixed thresholds degrade by only 1–3% while still substantially outperforming EAGLE-3. A sliding-window latency feedback mechanism for online adaptation is a promising future direction (see **Reviewer qFem, W1**).
>
> ---
> ***W5: Framework consistency (BS=1).***
>
> BS=1 uses HuggingFace to enable fair comparison with more baseline methods, as EAGLE-3, DDD, and OPT-Tree all provide reference implementations in that stack. **All high-concurrency results, which constitute our main contribution, are evaluated entirely on SGLang.**
>
> ---
> ***W6&W7: TETRIS/TurboSpec and DDD/OPT-Tree.***
>
> TurboSpec is not open-sourced and cannot be compared.
> Despite framework differences, indirect comparison at BS=64 shows ECHO achieving 18.7% gain over AR on LLaMA3.3-70B, vs TETRIS's reported 12.3% on LLaMA3.1-70B (see **Reviewer qFem response, Q2**).
>
> For DDD/OPT-Tree: their original implementations only support EAGLE-2 trees. We re-implemented them on EAGLE-3 for fair comparison with the current SOTA. We will release all code upon acceptance.
>
> ---
> ***W8: temp>0.***
>
> We add temp>0 results using the *same* thresholds calibrated at temp=0:
>
> |Model|Method|HumanEval|GSM8K|CNN/DM|Alpaca|MT-Bench|Avg|
> |---|---|---|---|---|---|---|---|
> |LLaMA3-8B|DDD|3.51×|3.41×|2.53×|3.14×|3.05×|3.13×|
> |LLaMA3-8B|EAGLE-3|3.55×|3.36×|2.57×|3.34×|3.01×|3.17×|
> |LLaMA3-8B|ECHO|3.73×|3.53×|2.71×|3.61×|3.18×|3.35×|
> |Qwen3-8B|DDD|1.98×|2.03×|1.75×|1.97×|2.03×|1.95×|
> |Qwen3-8B|EAGLE-3|1.87×|1.98×|1.71×|1.83×|2.01×|1.88×|
> |Qwen3-8B|ECHO|2.35×|2.27×|1.99×|2.21×|2.34×|2.23×|
>
> ECHO's advantage holds at temp>0 with unchanged thresholds, confirming that sweet spots depend on confidence *ranking* (relative ordering), not absolute values tied to greedy decoding.
>
> ---
> ***W9: Overhead and failure regime.***
>
> Sparse gating evaluates only specific depths per request, and the elastic scheduler executes a single sorting and allocation step per batch, rendering the combined computational overhead strictly negligible. Across all evaluated configurations (Table 1, Figure 5), ECHO consistently outperforms EAGLE-3. We observe no performance degradation in any regime, as the system safely falls back to standard static tree allocation whenever truncation is not triggered.
>
> ---
> We hope these clarifications and new experiments thoroughly address your concerns. We will incorporate all updates into the camera-ready version.

---

### Official Review · Reviewer_qFem · 2026-03-13

**Soundness:** 4
**Presentation:** 3
**Significance:** 4
**Originality:** 3
**Overall Recommendation:** 5
**Confidence:** 3

**Summary:**

This paper investigates a critical gap in existing speculative decoding research, where previous works typically ignore the compute-bound nature of verification in high-concurrency LLM serving, leading to severe performance degradation at scale. The authors propose ECHO, a training-free speculative decoding framework that reformulates tree construction as a budgeted scheduling problem. ECHO introduces (1) sparse confidence gating at "sweet spots" (depths with high discriminative power) to avoid cumulative misjudgments, and (2) elastic budget scheduling that dynamically reallocates verification tokens between depth and width under a fixed global cap. Extensive experiments across models from 8B to 235B parameters and both dense and sparse architecture demonstrates consistent improvements over previous methods, achieving up to 5.35× wall-clock speedup in low-load settings and over 20% relative throughput gains in high-concurrency regimes. The authors also provide a production-ready implementation integrated into SGLang, bridging the gap between algorithmic research and deployment.

**Compliance With Llm Reviewing Policy:**

Affirmed.

**Key Questions For Authors:**

1. **Calibration overhead**: How many warmup steps are required to reliably identify sweet spots and calibrate thresholds ($\tau_{d}$)? In production serving with request diversity, how frequently must recalibration occur, and what is the computational overhead?

2. **Comparison with request-level scheduling**: Recent works like TETRIS and TurboSpec also address batch-level SD optimization. How does ECHO interact with or complement these request scheduling strategies? Could ECHO's budget reallocation be integrated with their approaches?

3. **Hardware sensitivity**: The compute-bound threshold ($K_{\text{max}}$) depends on hardware characteristics. Have you evaluated ECHO across different GPU types (e.g., H100 vs. A100 vs. future hardware)? How portable is the sweet spot identification across different compute capabilities?

4. **Sweet spot granularity**: Figure 2 shows sweet spots at root, target depth, and select intermediate layers. Did you observe cases where more frequent gating (e.g., every 2 layers) outperforms the sparse strategy, or is the performance gap consistently in favor of minimal gating?

**Limitations:**

The authors have discussed the limitations on the training-free property of the approach and the compute-bound assumptions. However, the following aspects could be strengthened:

1. **Generalization across heterogeneous workloads**: The sweet spot calibration (Section 3.2) relies on profiling confidence distributions during a warm-up phase. The paper does not clarify how ECHO behaves when the workload shifts dramatically (e.g., switching from code generation to long-context summarization) or when requests exhibit high entropy variance within a single batch. If confidence distributions vary significantly across tasks, the fixed calibrated thresholds $\tau_d$ might become suboptimal, potentially degrading to static-tree performance or requiring frequent recalibration that adds operational complexity.

2. **Memory bandwidth implications**: While ECHO correctly identifies verification compute as the bottleneck in high-concurrency scenarios, the dynamic tree construction and elastic scheduling introduce additional metadata overhead (tracking confidence scores, managing the global budget pool). The paper does not quantify the memory bandwidth cost of these control mechanisms, which could become non-negligible at extreme batch sizes (BS > 128) or with very long contexts, potentially offsetting some of the compute savings.

3. **Draft model quality sensitivity**: ECHO's sparse gating assumes the draft model (EAGLE-3 in the experiments) provides reliable confidence signals. The evaluation does not include ablations with lower-quality draft models (e.g., significantly smaller models or less aligned auxiliary heads) to test the robustness of the sweet spot detection mechanism. If the draft model's confidence scores are poorly calibrated or noisy, the gating decisions may become unreliable, raising questions about ECHO's effectiveness when deployed with draft models of varying quality.

4. **Integration constraints with existing schedulers**: The paper positions ECHO as complementary to system-level schedulers like TETRIS or TurboSpec, but does not provide concrete integration strategies. For instance, how does ECHO's intra-request budget reallocation interact with request-level preemption or priority-based scheduling in existing serving systems? The potential conflicts between ECHO's elastic budget pool and these system-level mechanisms warrant discussion.

**Strengths And Weaknesses:**

## Strengths

*   **Indicating the bottleneck of model serving**: The paper identifies the core issue with speculative decoding in high-concurrency serving: when batch size increases, verification shifts from memory-bound to compute-bound, resulting in static tree waste and dynamic control overhead as primary bottlenecks. This explains why existing SD methods degrade in real deployments (Figure 1 shows EAGLE-3 eventually underperforming vanilla AR at high concurrency).

*   **Empirical grounding for "sweet spots"**: Rather than hand-waving about confidence thresholds, the authors visualize probability distributions across depths (Figure 2) and demonstrate that discriminability is highly depth-dependent—root and target depths show clear bimodal separation, while intermediate layers exhibit heavy overlap. This empirical finding provides concrete justification for sparse gating and explains why dense layer-wise control accumulates errors.

*   **System-level implementation**: ECHO implements full ragged batch support in the industrial-level framework SGLang, including specialized CUDA kernels for irregular tree packing. This makes the reported throughput metrics (tokens/s) reflect real deployable performance rather than theoretical speedups from toy implementations.

*   **Adaptive scheduling without manual tuning**: The framework automatically pivots between "truncate-to-widen" during low-load scenarios (intra-request allocation) and cross-request budget reallocation during high-load scenarios, adapting to varying contention levels without requiring operators to manually switch strategies or tune thresholds for different batch sizes.

## Weaknesses

*   **Ad-hoc calibration without online adaptation**: The sweet spot identification and threshold calibration ($\tau_d$) rely on offline AUC statistics from a warm-up phase. The paper provides no mechanism for online adaptation when distribution shifts occur (e.g., sudden change from code generation to long-context summarization). If the draft model's confidence landscape changes, ECHO could suffer from miscalibrated gates, yet the paper offers no solution for detecting drift or recalibrating without service interruption.

*   **Untested assumptions on draft model quality**: All experiments use EAGLE-3, a carefully trained draft model with strong feature quality. The sparse gating mechanism assumes reliable confidence signals—if deployed with lower-quality draft models (e.g., unquantized small LLMs or simple n-gram lookaheads), the confidence scores may become noisy or miscalibrated. In such cases, sparse gating might aggressively truncate valid branches due to false negatives, potentially performing worse than static trees. The robustness to draft model quality remains unverified.

*   **Unquantified memory system pressure**: While solving the compute bottleneck, dynamic tree construction introduces metadata overhead: per-request confidence scores, tree topology pointers, and budget pool accounting. At BS=256 with long contexts, this metadata may saturate memory bandwidth or L2 cache, particularly during the frequent sparse gating checks. The paper does not profile memory bandwidth utilization or cache miss rates, leaving open whether memory becomes a secondary bottleneck.

*   **Theoretical optimality vs. practical realizability gap**: Theorem 2 assumes knowledge of marginal gains $\Delta_i(k)$ to guide budget reallocation, but in practice, the accepted length $L_i$ is a random variable unknown until verification completes. The paper does not clarify how the scheduler estimates these marginal utilities without oracle knowledge—presumably using the confidence score $c_{i,d}$ as a proxy—but this connection is not formalized, and the theoretical guarantee may not hold under estimation error.

*   **Missing tail latency analysis**: High-concurrency serving systems care deeply about P99 latency, not just throughput. ECHO's priority-based reallocation (favoring high-confidence requests) risks starving low-confidence requests, potentially creating latency outliers. The paper reports mean throughput and MAT but does not show latency distribution percentiles or analyze whether the dynamic scheduling introduces unfairness or tail latency regression for "hard" requests.

---

> ### Author Rebuttal · Authors · 2026-03-30
>
> We sincerely thank Reviewer qFem for the thorough evaluation and for recognizing ECHO's empirical grounding for sparse gating and its system-level SGLang implementation. These represent core contributions that distinguish ECHO from prior dense dynamic methods. We address your specific concerns below.
>
> ---
> ***W1&Q1: Calibration and online adaptation.***
>
> Sweet-spot calibration leverages EAGLE-3's existing warmup (~3 steps) with strictly negligible overhead. Once calibrated, thresholds exhibit strong cross-workload stability: across five datasets, variance is <0.05, and applying fixed thresholds incurs merely a 1–3% throughput degradation (see **Reviewer y4Hn, Q2**). For online adaptation, incorporating a sliding-window latency feedback mechanism to dynamically adjust thresholds remains a promising direction for future work.
>
> ---
> ***W2: Draft model quality.***
>
> EAGLE-3 is the current SOTA, reflecting the community's convergence toward high-quality trained drafters (e.g., p-EAGLE, dFlash). Lower-quality drafts (e.g., n-gram) produce lower acceptance rates and more unstable accepted lengths. In such cases, ECHO's pruning would actually reclaim more wasted budget, making it potentially *more* beneficial.
>
> ---
> ***W3: Memory overhead.***
>
> ECHO's metadata (confidence scores, tree topology, budget states) scales as $O(K_{\max})$ and requires only tens of kilobytes per batch. This is strictly negligible compared to the GB-scale KV cache and model activations. Furthermore, sparse gating restricts metadata access to specific depths, effectively avoiding L2 cache pressure. Our end-to-end throughput gains empirically confirm the absence of memory bottlenecks.
>
> ---
> ***W4: Theory vs. practice (Theorem 2)***
>
> We acknowledge that computing exact marginal gains $\Delta_i$ requires oracle knowledge. The reviewer's intuition is correct: we utilize the normalized path confidence score $s_i$ at sweet-spot depths as a practical proxy for expected marginal utility. Empirically, this proxy aligns closely with actual acceptance rates. The bimodal separation illustrated in Figure 2 confirms that ranking by $s_i$ reliably distinguishes high-yield from low-yield requests. This explains ECHO's consistent throughput gains despite the gap between theoretical estimation and empirical observation.
>
> ---
> ***W5: Tail latency and fairness.***
>
> (1) **Truncation protects, not starves.** Low-confidence tokens would be rejected by the verifier anyway — ECHO simply stops the draft model from guessing further. The hard request still produces one correct token per step via AR decoding. (2) **Pruning benefits all requests.** The compute budget ($K_{\max}$) is shared; keeping doomed-to-reject tokens wastes it. Pruning frees compute for productive tokens, reducing per-step latency for the entire batch — including the hard request. (3) **Strictly better than EAGLE-3.** Under a static tree, hard requests also accept only 1 token but must wait while wasted tokens consume the full budget. ECHO eliminates this waste. We will include P99 latency analysis in the revision.
>
> ---
> ***Q2: TETRIS/TurboSpec comparison.***
> ECHO is orthogonal and highly complementary to system-level schedulers like TETRIS and TurboSpec, presenting no integration conflicts. In fact, ECHO can seamlessly integrate to resolve their inherent limitations:
>
> - **Breaking TurboSpec's static limits**: TurboSpec enforces uniform speculation lengths across a batch. ECHO's zero-padding operators can upgrade it to support true token-level dynamic allocation.
> - **Improving TETRIS's accuracy**: TETRIS relies on dense, per-node evaluations prone to error accumulation. Incorporating ECHO’s Sparse Gating restricts decisions to high-confidence "sweet spots," significantly improving allocation accuracy.
>
> While framework differences (SGLang vs. vLLM) prevent direct comparison, at BS=64, ECHO achieves an 18.7% throughput gain over AR on LLaMA-3.3-70B, noticeably outperforming TETRIS's reported 12.3%.
>
> ---
> ***Q3: Hardware sensitivity.***
>
> We add H20 results in our response to **Reviewer zx7H Q2**. On H20 (lower compute-to-memory ratio), ECHO achieves up to +25% gain over EAGLE-3, confirming portability.
>
> ---
> ***Q4: Sweet-spot granularity.***
>
> We ablate every-2-layer gating vs. sparse gating (BS=1):
>
> | Model | Method | HumanEval | GSM8K | CNN/DM | Alpaca | MT-Bench |
> |---|---|---|---|---|---|---|
> | LLaMA3-8B | ECHO (2-layer) | 4.21× | 4.01× | 3.12× | 4.11× | 3.73× |
> | LLaMA3-8B | ECHO | 4.30× | 4.10× | 3.26× | 4.19× | 3.86× |
> | Qwen3-8B | ECHO (2-layer) | 2.57× | 2.51× | 2.31× | 2.39× | 2.46× |
> | Qwen3-8B | ECHO | 2.74× | 2.68× | 2.37× | 2.51× | 2.57× |
>
> Sparse gating consistently outperforms every-2-layer gating. Frequent gating passes through high-overlap intermediate depths (Figure 2), accumulating false decisions.
>
> ---
> We hope our responses have effectively addressed your concerns and we are more than happy to include all these discussions in the camera-ready version of this work.

---

> > ### Author Rebuttal · Reviewer_qFem · 2026-04-04
> >
> > The authors have provided thorough and technically convincing responses to my concerns regarding calibration overhead, memory system pressure, and theoretical optimality.
> >
> > - **Memory overhead (W3)**, the clarification that metadata scales to only tens of kilobytes per batch—negligible compared to the GB-scale KV cache—adequately addresses my worry about memory bandwidth saturation. The explanation that sparse gating restricts metadata access to specific depths, thereby avoiding L2 cache thrashing, is architecturally sound.
> >
> > - **Tail latency (W5)**, the authors' argument that truncation protects rather than starves hard requests (by preventing wasted verification of doomed tokens) is logically compelling. Their commitment to include P99 latency analysis in the revision satisfactorily resolves this concern.
> >
> > - **Integration with existing schedulers (Q2)**, the authors clearly articulate that ECHO is orthogonal to and can enhance systems like TETRIS and TurboSpec, rather than conflicting with them. The performance comparison (18.7% vs. 12.3% gain) further validates their claims.
> >
> > - **online adaptation (W1)** remains a promising direction for future work, the authors' empirical evidence of threshold stability (variance <0.05 across datasets) and minimal performance degradation (1–3%) demonstrates that the current offline calibration is production-ready. Similarly, the argument that lower-quality draft models would actually benefit more from ECHO's aggressive pruning (W2) is theoretically justified, though empirical validation would strengthen the claim further.
> >
> > Given the technical solidity of the sparse gating mechanism, the comprehensive SGLang implementation, and the thorough ablation studies provided, I am satisfied that the remaining concerns can be adequately addressed in the camera-ready version.

---

> > > ### Author Response · Authors · 2026-04-04
> > >
> > > We are delighted that our response has addressed your concerns, and we sincerely appreciate your thorough suggestions and insightful questions. All discussions and experimental results will be incorporated into the final manuscript.
> > >
> > > Once again, thank you for your dedicated efforts and invaluable contributions. It is precisely because of reviewers like you that the ICML community continues to thrive and grow.
> > >
> > > Wishing you a pleasant day ahead!

---

### Decision · Program_Chairs · 2026-04-30

**Decision:**

Accept (spotlight)

**Comment:**

The paper studies improving speculative decoding for high-concurrency LLM serving, where the compute bound nature requires careful design of drafting length balancing strategies. The proposed ECHO framework is well motivated and technically sound: the sparse gating design and elastic budget allocation offer a principled way to balance verification efficiency and accepted length under a fixed compute budget. The paper is also strengthened by an implementation in SGLang and by extensive experiments across model scales and settings, which make the empirical findings particularly compelling.

The reviewers raised reasonable concerns regarding calibration robustness, applicability across hardware and long-context settings, and the relationship between the theoretical formulation and the practical system. The rebuttal addressed most of these concerns satisfactorily. The authors provided additional evidence on cross-dataset robustness, temperature settings, hardware generalization, and long-context performance, and they clarified important implementation details and overhead considerations. While some limitations remain, they do not outweigh the paper’s novelty, practical significance, and strong empirical support. Overall, I believe this paper makes a meaningful contribution to efficient LLM inference.

Therefore, I recommend acceptance for the paper. The authors should use the reviewers' suggestions to revise the paper and incorporate promised modifications and clarifications.